# Coordination of Pickpocket ion channel delivery and dendrite growth in Drosophila sensory neurons

Josephine W. Mitchell[1,2,3], Ipek Midillioglu[4ʘ], Ethan Schauer[4ʘ], Bei Wang[5,6], Chun Han[5,6], Jill Wildonger[2,4,7] *

1 Integrated Program in Biochemistry, University of Wisconsin-Madison, Madison, Wisconsin, United States of America, 2 Biochemistry Department, University of Wisconsin-Madison, Madison, Wisconsin, United States of America, 3 Department of Chemistry and Biochemistry, Kalamazoo College, Kalamazoo, Michigan, United States of America, 4 Pediatrics, University of California, San Diego, La Jolla, California, United States of America, 5 Weill Institute for Cell and Molecular Biology, Cornell University, Ithaca, New York, United States of America, 6 Department of Molecular Biology and Genetics, Cornell University, Ithaca, New York, United States of America, 7 Cell & Developmental Biology, School of Biological Sciences, University of California, San Diego, La Jolla, California, United States of America

ʘ These authors contributed equally to this work.
* jwildonger@ucsd.edu

## Abstract

Sensory neurons enable an organism to perceive external stimuli, which is essential for survival. The sensory capacity of a neuron depends on the elaboration of its dendritic arbor and the localization of sensory ion channels to the dendritic membrane. However, it is not well understood when and how ion channels localize to growing sensory dendrites and whether their delivery is coordinated with growth of the dendritic arbor. We investigated the localization of the DEG/ENaC/ASIC ion channel Pickpocket (Ppk) in the peripheral sensory neurons of developing fruit flies. We used CRISPR-Cas9 genome engineering approaches to tag endogenous Ppk1 and visualize it live, including monitoring Ppk1 membrane localization via a novel secreted split-GFP approach. Fluorescently tagged endogenous Ppk1 localizes to dendrites, as previously reported, and, unexpectedly, to axons and axon terminals. In dendrites, Ppk1 is present throughout actively growing dendrite branches and is stably integrated into the neuronal cell membrane during the expansive growth of the arbor. Although Ppk channels are dispensable for dendrite growth, we found that an over-active channel mutant severely reduces dendrite growth, likely by acting at an internal membrane and not the dendritic membrane. Our data reveal that the molecular motor dynein and recycling endosome GTPase Rab11 are needed for the proper trafficking of Ppk1 to dendrites. Based on our data, we propose that Ppk channel transport is coordinated with dendrite morphogenesis, which ensures proper ion channel density and distribution in sensory dendrites.

**Data Availability Statement:** All data are in the manuscript and/or supporting information files.

**Funding:** This work is supported by National Institute of Neurological Disorders and Stroke

(https://www.ninds.nih.gov/) grants R01NS102385 (J.W.) and R01NS099125 (C.H.). The funder had no role in study design, data collection and analysis, decision to publish, or preparation of the manuscript.

**Competing interests:** The authors have declared that no competing interests exist.

## Author summary

Peripheral sensory neurons are essential for an organism to interact with its environment. Neurons are composed of signal-receiving dendrites and a signal-sending axon. Ion channels distributed throughout sensory dendrites transduce external stimuli into chemical signals, however the mechanisms that localize ion channels to sensory dendrites are not well understood. Both the composition of ion channels in the dendrites and the structure of a sensory neuron's dendritic arbor are important for how it functions to sense external stimuli. Using live imaging and genomic engineering, we have discovered that the localization of a sensory ion channel, Pickpocket, in fruit fly sensory neurons is coordinated with growth of the dendritic arbor and that Pickpocket levels scale in proportion to dendrite length, even when transport to dendrites is disrupted. We also developed a novel genetically encoded approach to visualize the membrane expression of proteins in a living organism utilizing the split-GFP system. We found that both the secretory and endosomal networks mediate the forward trafficking of Pickpocket during neuronal morphogenesis, thus coordinating membrane growth with ion channel delivery. Our findings reveal that actively growing sensory dendrites are equipped with the ion channels needed for sensing external stimuli.

## Introduction

An organism's interactions with its environment rely on its ability to detect external stimuli through sensory neurons. Ion channels distributed throughout the dendritic arbor of a sensory neuron rapidly transduce external stimuli into cellular signals. Both the morphology of a sensory neuron's dendritic arbor and the localization of ion channels in the arbor are essential for the establishment of a neuron's receptive field and sensory capacity. While the localization of ion channels to synapses in the central nervous system has been well studied [1], little is known regarding mechanisms that regulate the delivery of ion channels to the dendritic membrane of sensory neurons in the peripheral nervous system. It is also not known whether and how this trafficking may be coordinated with dendrite morphogenesis and the expansion of the dendritic arbor to establish the proper distribution of ion channels needed for sensing environmental stimuli.

To investigate the relationship between ion channel trafficking and dendrite growth, we used the *Drosophila melanogaster* class IV dendritic arborization (da) neurons as a model. The class IV da neurons function as polymodal nociceptors that detect multiple stimuli (thermal, mechanical, and light) and extend elaborately branched dendritic arbors that cover the larval body wall [2–5]. These neurons are an ideal model to study ion channel delivery in growing sensory dendrites for several reasons. First, during larval development, the class IV da neuron dendrites undergo expansive growth that can be easily visualized live in intact animals due to their superficial location just beneath the transparent larval cuticle and their relatively flat, two-dimensional morphology [6,7]. Second, the class IV da neurons have been a powerful *in vivo* model to identify mechanisms of dendrite morphogenesis, including players involved in membrane production and trafficking, the secretory and endosomal networks, molecular motor-based transport, and the cytoskeleton [8,9]. By manipulating known mechanisms of dendrite arbor growth, we can investigate how ion channel trafficking is coordinated with dendrite morphogenesis. Third, the general morphology and function of the class IV neurons is similar to peripheral sensory neurons and nociceptors in other organisms, including the mammalian C- and Aδ-fibers and the worm PVD and FLP neurons [10–12]. Thus, studying ion

channel trafficking in the class IV da neurons may shed light on conserved mechanisms of ion channel localization in sensory dendrites.

During neuronal morphogenesis, the class IV da neurons express several dendritic ion channels that have been structurally and functionally characterized, including Pickpocket (Ppk); Transient Receptor Potential (TRP) channels, such as TrpA1 and painless; and Piezo [3,13–17]. Whereas TRP and Piezo channels are comprised of large multi-pass membrane protein subunits, the Ppk ion channel subunits are relatively small, two-pass membrane proteins. Their modest size makes endogenous Ppk channels amenable to manipulation via CRISPR-Cas9 genome engineering. Moreover, the crystal structure of a conserved Ppk ortholog in chickens, ASIC1, has been solved, providing information that can be leveraged for the structure-guided manipulation of Ppk [18]. For these reasons we decided to focus on investigating the trafficking of Ppk in the class IV da neurons.

Ppk proteins belong to the large, structurally conserved family of Degenerin/Epithelial Na$^+$ Channel/Acid Sensing Ion Channels (DEG/ENaC/ASICs) whose members in worms, flies, fish, and mammals carry out a variety of functions ranging from mechanosensation and learning and memory in the nervous system to salt homeostasis in epithelial cells in the kidney [19–21]. In the fly class IV da neurons, the Ppk channel is composed of two subunits, Pickpocket 1 (Ppk1) and Pickpocket 26 (Ppk26), which are mutually dependent on each other for membrane expression [16,22–24]. The localization of Ppk1 and Ppk26 has been characterized using antibodies and fluorescently tagged transgenes, and both subunits are broadly distributed throughout the developing dendrites of class IV da neurons. Interestingly, Ppk1 and Ppk26 are expressed from late-embryo to mid-larval stages, which coincides with the initiation of dendrite development and the period of expansive growth as the dendrite arbor increases in size over 100-fold [6,16,22,25–27]. The timing of Ppk1 and Ppk26 expression suggests that Ppk channel production and localization is coordinated with dendrite morphogenesis.

To investigate the coordination of Ppk1 trafficking and dendrite growth, we used CRISPR-Cas9 genome engineering to tag endogenous Ppk1 and follow its localization in growing dendrites. We found that in developing neurons, Ppk1 is enriched throughout dendrites and, unexpectedly, is also present in axons and axon terminals. To visualize the membrane localization of Ppk1, we developed a new secreted split-GFP-based strategy to monitor the insertion of proteins into the cell membrane live in developing neurons and other cells in vivo. This approach revealed that Ppk1 was present throughout the dendritic arbor and actively extending branches, which suggests that Ppk1 is integrated into the nascent membrane that is added to growing dendrites. Consistent with this idea, we found that Ppk1 density scales in proportion to dendrite arbor size. Our data also implicate the molecular motor dynein and endosomal GTPase Rab11 in the localization of Ppk1 to dendrites. Combined, our results suggest a model in which sensory neurons regulate dendritic ion channel density by packaging ion channels (e.g., Ppk) in the membrane that grows and expands the dendritic arbor.

## Results

### Ppk1 is present in both the dendrites and axons of developing peripheral sensory neurons

To visualize the localization of the Ppk ion channel in developing neurons, we tagged endogenous Ppk1 with fluorescent proteins. To facilitate the manipulation of *ppk1*, we first replaced the ppk1 gene with an attP "docking site," which enables the reliable and rapid knock-in of new *ppk1* alleles (Fig 1A and 1B). We then used this strain to knock-in *ppk1* tagged with one copy of superfolder GFP (sfGFP) at either the N- or C-terminus (Figs 1C and S1; since Ppk1 tagged with GFP at either terminus displayed similar localization, we used C-terminally tagged

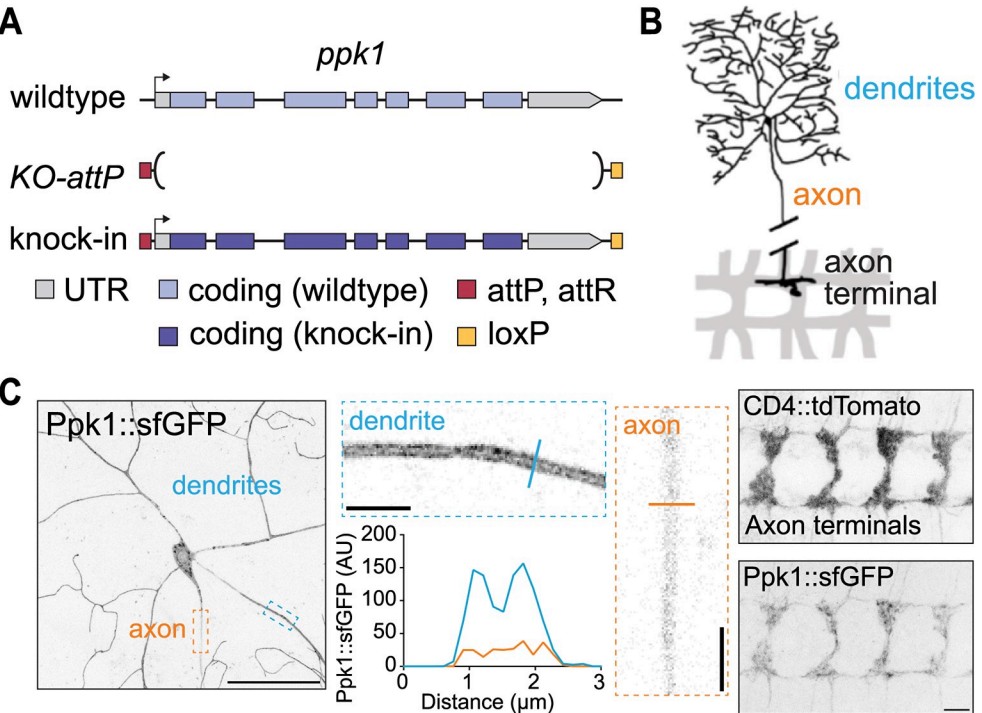

**Fig 1. Fluorescently tagged endogenous Ppk1 localizes to dendrites and axons.** (A) Cartoon illustrating the CRISPR-Cas9-engineered *ppk1* locus in which endogenous *ppk1* was replaced with an attP "docking" site for reliable, rapid knock-in of new *ppk1* alleles. (B) Cartoon of a single class IV ddaC neuron. The ddaC dendrites cover the larval body wall in the periphery and extend a single unbranched axon into the ventral nerve cord (VNC). There are three *ppk1*-expressing class IV da neurons per hemisegment, and the axon terminals of the class IV da neurons form a railroad track-like pattern in the VNC. (C) Representative images of Ppk1::sfGFP in the cell body, dendrites, and axon in a ddaC neuron in the periphery of a live 3rd instar larva (left), and Ppk1::sfGFP in the class IV da neuron axon terminals of a fixed VNC (right). Dashed-outline boxes: Individual 1-µm thick z-plane zoomed-in views of dendrites and axons; a line indicates the position at which an intensity profile plot was generated. CD4::tdTomato marks the axon terminals of the class IV neurons in the VNC. Scale bars, 50 µm (peripheral ddaC neuron image, left), 10 µm (VNC image, right), and 5 µm (dashed-outline boxes).

Ppk1 for our experiments). We observed robust fluorescent signal in neurons in live animals with just one copy of GFP attached to Ppk1, indicating that GFP-tagged Ppk1 is an advantageous tool to monitor endogenous channel localization (Fig 1C). Consistent with previous reports, Ppk1::sfGFP was expressed in the class IV da neurons in the peripheral nervous system [2,16,26]. In the dorsal class IV da neuron called ddaC, Ppk1::sfGFP localized to both dendrites and axons, but its distribution to and within these compartments differed. Ppk1::sfGFP was enriched in dendrites, where it appeared to localize predominantly to the dendritic membrane and was present throughout the dendritic arbor (Fig 1C). This distribution matches the previously reported distribution of Ppk1 based on antibody staining and fluorescently tagged Ppk1 transgenes [22,24,28]. In contrast to dendrites, the Ppk1::sfGFP signal was dimmer in axons and did not appear to align with the axonal membrane (Fig 1C). In the ventral nerve cord, where the ddaC axons terminate, Ppk1::sfGFP was present in axon terminals (Fig 1C). Altogether, our data indicate that Ppk1::sfGFP localizes predominantly to dendrites but is also present in axons and axon terminals.

In addition to tagging Ppk1 with sfGFP, we tagged Ppk1 with mCherry. The distribution of Ppk1::mCherry was similar to Ppk1::sfGFP; however, unlike Ppk1::sfGFP, Ppk1::mCherry clustered in bright puncta in the cell body, proximal dendrites, and axon (S2A and S2B Fig). The

Ppk1 partner subunit Ppk26 displays a similar punctate distribution pattern when tagged with the photoconvertible fluorescent protein Dendra2, but GFP-tagged Ppk26 resembles Ppk1::sfGFP [24,29]. The distribution pattern of Ppk1 tagged with both sfGFP and mCherry was similar to the singly tagged Ppk1 proteins (S2B and S2C Fig), which suggests that the bright punctate mCherry signal may reflect mCherry fluorescence in a compartment where sfGFP fluorescence is quenched. The Ppk1::mCherry puncta colocalized with Rab5 and Rab7 and were disrupted by a dominant-negative Rab5, which suggests that Ppk1::mCherry may illuminate Ppk1 in a degradative pathway and organelle(s) whose conditions are not optimal for sfGFP fluorescence (S2D and S2E Fig) [30]. As the brightness of the mCherry puncta impacted our imaging of the non-punctate mCherry signal, we used GFP-tagged Ppk1 for the majority of our experiments.

## Novel split-GFP-based approach to monitor the neuronal cell membrane localization of Ppk1

Fluorescently tagged Ppk1 reveals the overall localization of Ppk1, but we also wanted to track where Ppk1 was inserted into the neuronal cell membrane. To monitor the membrane localization of Ppk1, we initially tagged Ppk1 with superecliptic pHluorin. Superecliptic pHluorin (referred to simply as pHluorin hereafter) is a pH-sensitive GFP variant that is used to monitor the insertion of transmembrane proteins into the cell membrane because it fluoresces in neutral pH environments, such as extracellular space, but has minimal fluorescence in low pH environments, such as the lumen of a transport vesicle [31]. To determine an optimal extracellular position at which to add a fluorescent tag such as pHluorin, we first tagged an extracellular (EC) loop of endogenous Ppk1 with sfGFP and compared its fluorescence to Ppk1 tagged with sfGFP at the N- or C-terminus. We tested two positions: Site 1 was selected based on the structure of chicken ASIC1, and Site 2 is near a position that was previously used to monitor rat ASIC1a membrane insertion via a haemagglutinin epitope tag [32] (S3 Fig). We found that insertion of sfGFP at Site 1 resulted in fluorescence similar to Ppk1::sfGFP, whereas the insertion of sfGFP at Site 2 led to relatively weak fluorescence (S3A Fig). We then tagged Ppk1 with pHluorin at Site 1 and found that Ppk1::pHluorin$^{EC}$ produced relatively weak fluorescence (S3B Fig). We next tested the effects of eliminating Ppk26 on Ppk1::pHluorin$^{EC}$ fluorescence and distribution, as the loss of Ppk26 interferes with the membrane localization of Ppk1 [22–24]. In *ppk26* mutant neurons, Ppk1::pHluorin$^{EC}$ fluorescence noticeably increased in the cell body and decreased in dendrites, but fluorescence was still clearly visible in patches in dendrites, particularly at dendrite branch points (S3C Fig). Given that the membrane localization of Ppk1 depends on Ppk26, the fluorescence pattern of Ppk1::pHluorin$^{EC}$ in the *ppk26* mutant neurons suggested that Ppk1::pHluorin$^{EC}$ might fluoresce in the neutral environment of the endoplasmic reticulum (ER) as well as in the cell membrane. The potential for Ppk1::pHluorin$^{EC}$ to fluoresce in both the ER and cell membrane significantly limits its use as a tool to monitor the insertion of Ppk1 into the neuronal membrane in vivo.

The weak fluorescent signal of Ppk1::pHluorin$^{EC}$, and its ability to fluoresce in the ER, led us to develop a new split-GFP-based approach to monitor the membrane insertion of endogenous Ppk1 (Fig 2A). This approach is analogous to other recently reported split-GFP techniques to monitor the localization of transmembrane proteins [33,34]. We tagged endogenous Ppk1 at Site 1 in an extracellular loop with three copies of the split-GFP peptide GFP(11). In addition to the extracellular GFP(11) tag, we also tagged the Ppk1 C-terminus with mCherry, which enabled us to follow Ppk1 localization throughout the neuron (Ppk1::GFP(11x3)$^{EC}$::mCherry$^{C-term}$). We then expressed a secreted version of GFP(1–10) (secGFP(1–10)) in fat cells, which released secGFP(1–10) into the hemolymph of the larval open circulatory system.

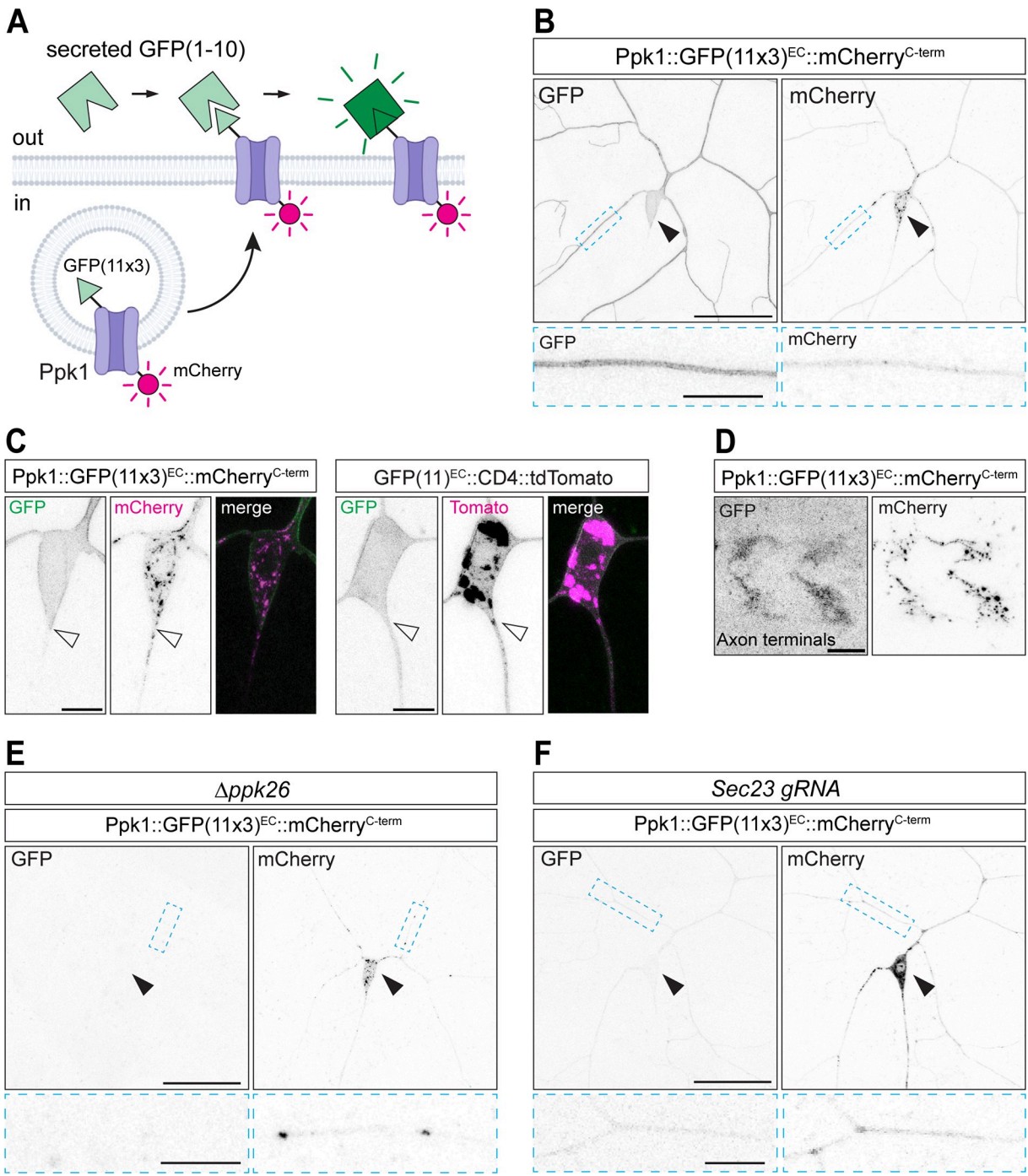

**Fig 2. Membrane localization of Ppk1 visualized in live larvae with the secreted split-GFP system.** Images of ddaC neurons in live 3rd instar larvae (B, C, E, F) and a fixed VNC (D). UAS-secGFP(1–10) expression was driven in the fat bodies by DcG-Gal4. (A) Cartoon of the secreted-split-GFP-based approach to label Ppk1 when it is inserted into the neuronal cell membrane. GFP(11) is positioned on an extracellular loop of Ppk1. Secreted-GFP(1–10) (secGFP(1–10)) is expressed by the fat bodies, which secrete secGFP(1–10) into the hemolymph that circulates throughout larvae. Neither GFP(11) nor secGFP(1–10) is fluorescent on its own. When Ppk1 inserts into the neuronal cell membrane, GFP(11) binds secGFP(1–10), resulting in reconstitution of GFP and fluorescent signal. (B) Representative image of GFP and mCherry fluorescence in the dendrites, cell body, and axon of a ddaC neuron expressing Ppk1 dual-tagged with 3 copies of GFP(11) on an extracellular (EC) loop and mCherry at the C-terminus (Ppk1::GFP(11x3)EC::mCherryC-term). (C) Representative images of GFP and a red fluorescent protein (mCherry or tdTomato) in ddaC neurons expressing either Ppk1::GFP(11x3)EC::mCherryC-term (left) or CD4 tagged with one copy of GFP(11) at the extracellular N-terminus and with tdTomato at the intracellular C-terminus (GFP(11)EC::CD4::tdTomato). (D) Representative image of Ppk1::GFP(11x3)EC::mCherryC-term in class IV da neuron axon terminals in the VNC. (E, F) Loss of *ppk26* (Δppk26: *ppk26*^Δ11/Δ11) (E) or the targeted

disruption of Sec23 in class IV da neurons (*ppk-Cas9 U6:3-Sec23-gRNA*) (F) affects the GFP fluorescence of Ppk1::GFP(11x3)^EC::mCherry^C-term. Solid arrowheads: cell body; open arrowheads: cell body-axon boundary. Scale bars, 50 μm (solid-outline panels in B, E, F,), 10 μm (dashed-outline boxes in B, E, F and solid-outline panels, D), and 5 μm (C).

Thus, Ppk1 tagged with GFP(11) in an extracellular loop will only fluoresce when Ppk1 is inserted into the cell membrane and encounters secGFP(1–10) (Figs 2A and S4A). As a control, we used a construct in which the extracellular N-terminus of the single-pass transmembrane protein CD4 is tagged with GFP(11) and the intracellular C-terminus is tagged with tdTomato (GFP(11)^EC::CD4::tdTomato) (S4B Fig) [35].

We monitored GFP fluorescence from Ppk1 and CD4 tagged with GFP(11). Both Ppk1::GFP(11x3)^EC::mCherry^C-term and GFP(11)^EC::CD4::tdTomato exhibited GFP fluorescence in dendrites and cell bodies (Figs 2B and S4B). In neurons expressing Ppk1::GFP(11x3)^EC::mCherry^C-term, we observed a distinct boundary of GFP fluorescence between the cell body and proximal axon, the latter of which was devoid of fluorescent signal (Fig 2C). While the proximal axon and axon shaft lacked GFP fluorescence, we observed GFP fluorescence in the axon terminals of neurons expressing Ppk1::GFP(11x3)^EC::mCherry^C-term (Fig 2D). In contrast to Ppk1 tagged with GFP(11), CD4 tagged with GFP(11) displayed GFP fluorescence throughout the proximal axon and axon shaft (Fig 2C).

We then took two approaches to determine whether the GFP fluorescence of Ppk1 tagged with GFP(11x3) reflected membrane-localized Ppk1. As described above, the membrane localization of Ppk1 depends on Ppk26 [22–24]. In neurons lacking Ppk26, there was no GFP fluorescent signal from Ppk1::GFP(11x3)^EC::mCherry^C-term, albeit the mCherry signal was still visible, indicating that Ppk1 was still produced in the absence of Ppk26 (Fig 2E). Next, we disrupted the membrane localization of Ppk1 by targeting Sec23, an essential component of the COPII complex that promotes the budding of cargo-containing vesicles from the ER. We used a transgenic guide RNA (gRNA) that targets the first exon of *Sec23* in combination with a transgene that expresses Cas9 specifically in the class IV da neurons (this limits the disruption of *Sec23* to the class IV da neurons and thus the secretory pathway in other cells is not affected) [36]. We found that *Sec23* loss-of-function neurons had virtually no GFP fluorescence, consistent with the idea that Ppk1 tagged with GFP(11) reports the membrane localization of Ppk1 (Fig 2F). Thus, we have generated a new tool to monitor the live membrane localization of endogenous Ppk1 in intact larvae, and this approach can also be applied to reveal the cell membrane localization of other transmembrane proteins in vivo.

## Ppk1 is present in the dendritic membrane as dendrite branches form and extend

The localization of Ppk1 and Ppk channels throughout the dendritic arbor and dendritic membrane raises the question of how Ppk1 becomes so broadly localized. One possibility is that Ppk1 is localized to dendrites at the same time that the arbor is created. Indeed, consistent with the previously reported onset of *ppk1* expression, we first observed Ppk1::sfGFP in cell bodies and proximal axons just prior to dendrite extension, and we subsequently observed Ppk1::sfGFP in newly formed dendrites (Fig 3A) [16,22,25–27]. Next, we asked whether Ppk1 is present in actively growing dendrites. Since imaging fluorescently tagged Ppk1 live in nascent dendrites during embryonic stages was technically challenging, we imaged fluorescently tagged Ppk1 in developing dendrites during larval stages. Ppk1::sfGFP and Ppk1::GFP(11x3)^EC were present throughout growing dendrites, in both branches that formed *de novo* and existing branches that extended (Fig 3B and 3C). We imaged Ppk1::sfGFP and Ppk1::GFP(11x3)^EC continuously and at intervals over a period of several minutes, and at both time scales

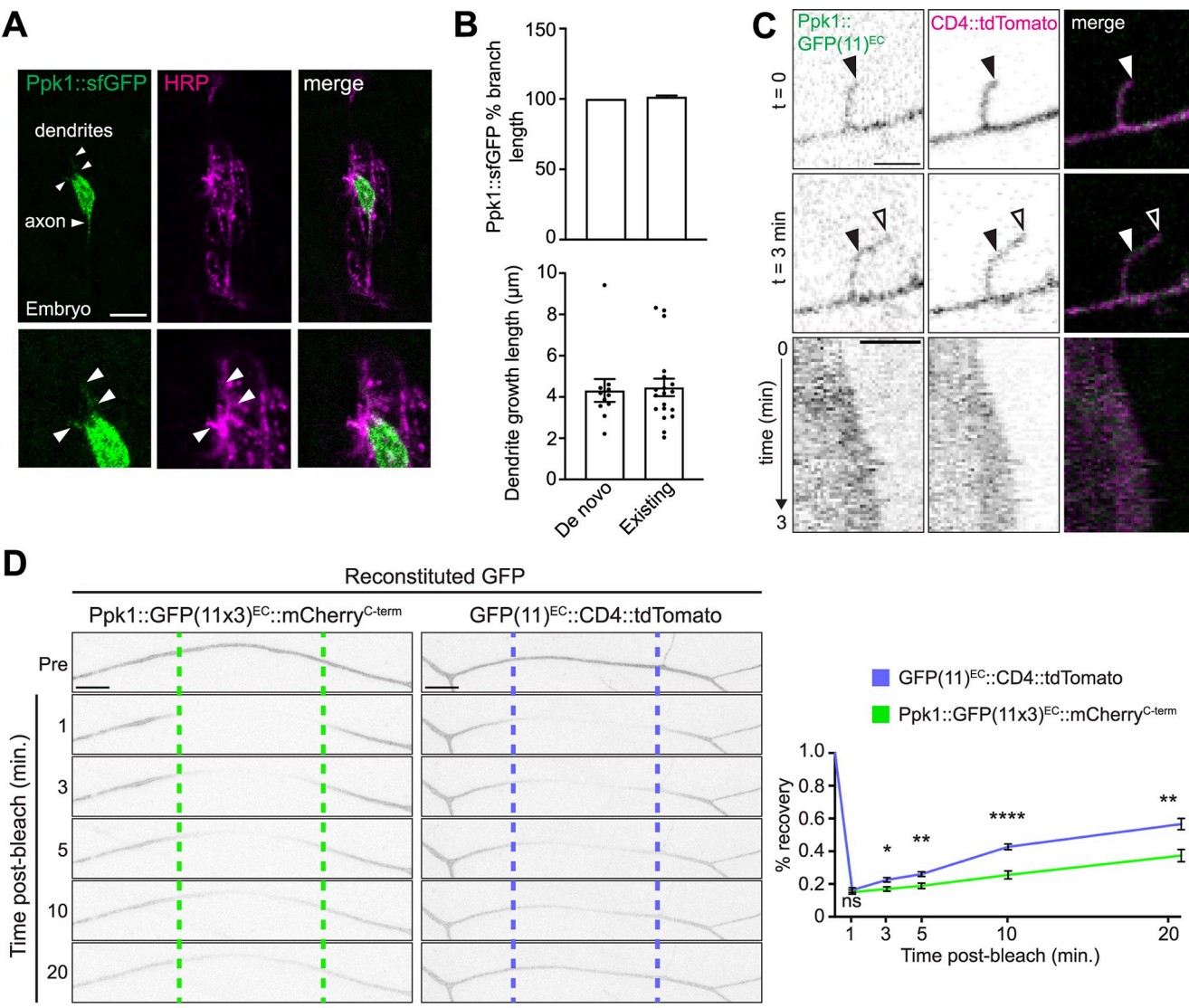

**Fig 3. Ppk1 is present in actively growing dendrites including dendrite tips.** Images of ddaC neurons in fixed embryos (A) and live 3rd instar larvae (B-D). (A) Ppk1::sfGFP in the cell body and axon of a ddaC neuron in a late-stage embryo. The ddaC neuron is part of a cluster of sensory neurons, which are marked by anti-HRP. Arrowheads point to Ppk1::sfGFP in nascent dendrites. Scale bar, 10 µm. (B) Quantification of the distribution of Ppk1::sfGFP in growing dendrite branches over 30 minutes. Top graph: In growing branches, Ppk1::sfGFP signal was quantified as a percentage of the branch length marked by CD4::tdTomato (de novo: 11 branches, 4 neurons; existing: 18 branches, 5 neurons). Ppk1::sfGFP was present along the entire length of the branch. Bottom graph: In 30 minutes, both newly formed and existing branches grew equivalent lengths (de novo: 11 branches, 4 neurons; existing: 19 branches, 5 neurons). In the graphs, each data point represents a dendrite branch. (C) Still images (top) and kymographs (bottom) from a 3-minute movie showing GFP fluorescence from Ppk1::GFP(11x3)EC in a growing dendrite tip marked by CD4::tdTomato. The start and end points of dendrite growth are indicated by solid and open arrowheads, respectively. Genotype: *Ppk1::GFP(11x3)EC/DcG-Gal4; UAS-secGFP(1–10)/ppk-CD4::tdTomato*. Scale bars, 3 µm. (D) Representative images and quantification of FRAP of GFP in neurons expressing Ppk1::GFP(11x3)EC::mCherryC-term and GFP(11)EC::CD4::tdTomato (both genotypes: 8 neurons, 8 larvae). In the image montage, dashed lines represent the bleached region, which was within 50–150 µm of the cell body. Quantification, percent recovery: Student's unpaired t-test, 1 min (p = 0.5417), 3 min (p = 0.0108), 5 min (p = 0.0064); 10 min (p<0.0001); 20 min (p = 0.0021). The GFP signal in a 10 µm section centered in the bleached region was quantified at each of the indicated time points. Scale bar, 10 µm. In the graph, each data point represents a neuron. n.s. = not significant (p>0.05), *p<0.05, **p = 0.01–0.001, and ****p<0.0001. In the graphs, the data are plotted as mean ± SEM (B, D).

we consistently observed Ppk1::sfGFP throughout dendrite branches, including dendrite tips, as they grew and, in some instances, retracted.

It is possible that the broad distribution of Ppk1 throughout the dendritic arbor and in growing dendrites might reflect its ability to readily diffuse in the dendritic membrane. To test

this we carried out fluorescence recovery after photobleaching (FRAP) with GFP(11)-tagged Ppk1 and, as a control, GFP(11)-tagged CD4. After photobleaching, GFP fluorescence from Ppk1::GFP(11x3)$^{EC}$::mCherry$^{C-term}$ recovered gradually over tens of minutes, significantly slower than the GFP(11)$^{EC}$::CD4::tdTomato control (Fig 3D). These FRAP results indicate that rapid diffusion of Ppk1 is unlikely to explain its broad distribution and presence in growing dendrites.

## Ppk1 density is not affected by decreasing dendrite arbor size but is reduced when dendrite length significantly increases

One potential model to explain the broad distribution of Ppk1 in the dendritic arbor is that Ppk1 is an integral component of the membrane that drives the expansive growth of the arbor. If the dendritic localization of Ppk1 is tightly coordinated with dendrite arbor growth, then Ppk1 density should be proportional to dendrite arbor size even when dendrite growth is altered. To test this idea, we took advantage of different conditions that are known to reduce or increase dendrite growth, acknowledging that these experiments could only serve as a general test of whether Ppk1 density scales with dendrite arbor size. For these experiments we analyzed the distribution of Ppk1::sfGFP since we were unable to assay Ppk1::GFP(11x3)$^{EC}$ due to technical difficulties. First, we tested two different conditions that reduce dendrite growth: knock-down of the ribosomal protein Rpl22, which disrupts the extensive elaboration of dendrite branches, and expression of a dominant-negative form of the ecdysone receptor (EcR-DN), which perturbs the overall expansive growth of the arbor [37,38]. Under both conditions, total dendrite length and number of terminal tips significantly decreased as expected, however the dendritic density of Ppk1::sfGFP was similar to controls (Figs 4A, 4B, and S5). We also assayed Ppk1 density when dendrite growth was enhanced via overexpression of either the actin regulator Rac1 [39] or phosphoinositide 3-kinase (PI3K), which regulates cell growth through the mTOR (mechanistic target of rapamycin) pathway [6]. As previously reported, Rac1 overexpression increased dendrite branch number without affecting dendrite length, and we found that Ppk1::sfGFP density was not affected (Figs 4C and S5). Overexpression of PI3K increased both dendrite branch number and dendrite length by approximately a quarter, and, in contrast to the other growth-perturbing conditions, Ppk1::sfGFP density was reduced by approximately a quarter (Figs 4D and S5). In this overgrowth condition it is possible that the production of Ppk1::sfGFP is outpaced by dendrite growth. Although this is only a small sampling of the many conditions that affect dendrite growth, our results, coupled with other reports of Ppk1 or Ppk26 density in mutants that affect arbor size size [40–42], indicate that Ppk density typically remains constant or is only minimally affected despite dramatic changes in dendrite arbor size.

Our finding that Ppk1 density remains constant despite changes in dendrite growth raises the question of whether other ion channels in sensory neuron dendrites might similarly scale in proportion to arbor size. The class IV da neurons also express the TRP channel TrpA1 [17]. To monitor TrpA1 localization and density, we used a protein-trap strain in which GFP is inserted into endogenous TrpA1 near the N-terminus (GFP::TrpA1). Unlike Ppk1::sfGFP, we found that GFP::TrpA1 density fluctuated when dendrite growth was perturbed by either EcR-DN or elevated Rac1 (S6 Fig). Thus, GFP::TrpA1 does not scale proportionally with changes in dendrite growth, which suggests that ion channels in dendrites are likely differentially regulated.

## Dendrite arborization disrupted by over-active mutant Ppk26

The presence of Ppk1 in the membrane of growing dendrites led us to investigate the potential role of Ppk channels in dendrite morphogenesis. In neurons lacking both Ppk1 and Ppk26,

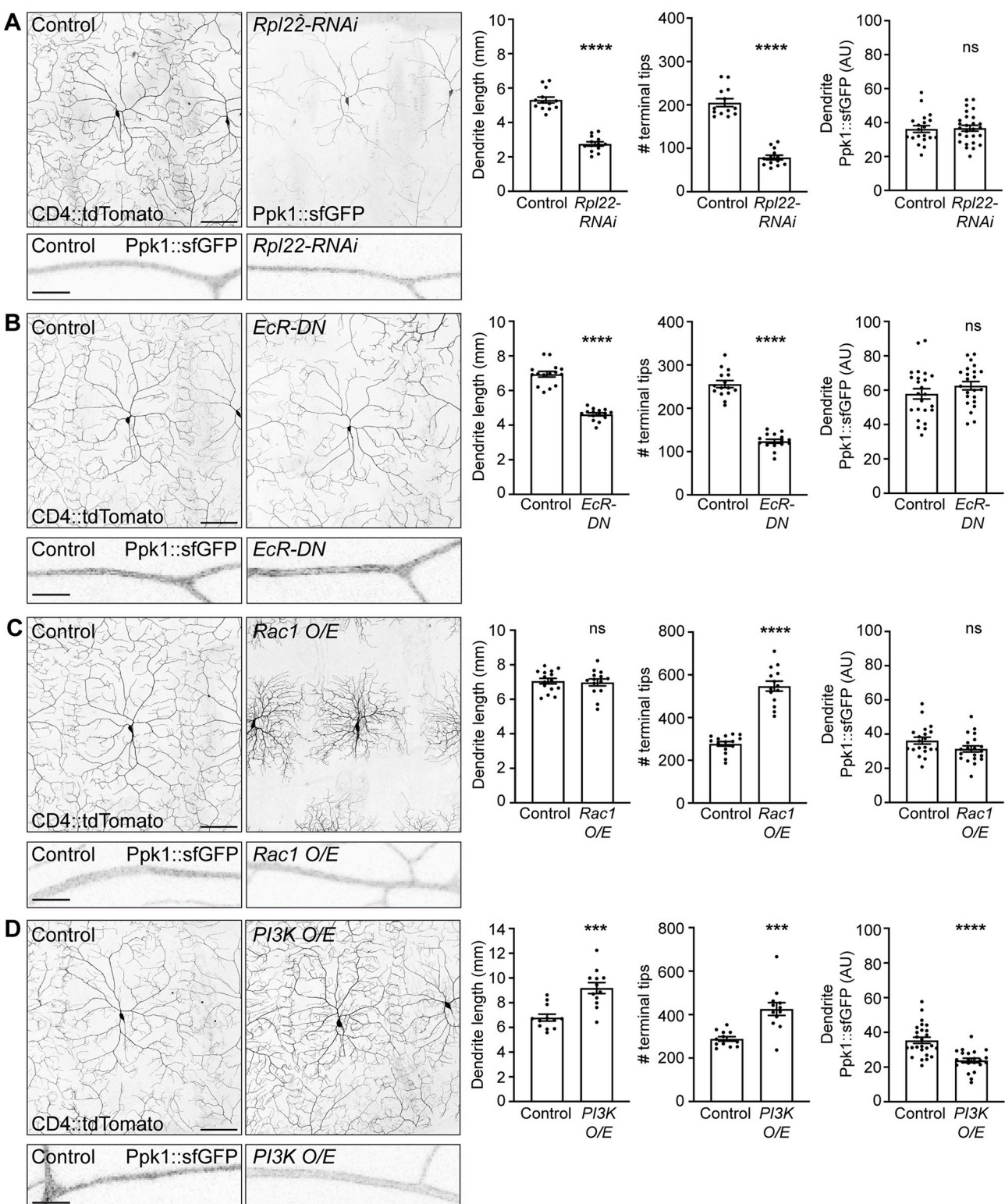

**Fig 4. Ppk1 levels are not affected by decreasing dendrite arbor size but are reduced when dendrite length increases.** Images of ddaC neuron morphology in live 2nd instar larvae (72 h AEL). The dendritic membrane is marked by CD4::tdTomato, unless otherwise noted. Zoomed inset images show Ppk1::sfGFP in dendrite segments of neurons in live 3rd instar larvae (120 h AEL). (A) Representative images and quantification of control and *Rpl22-RNAi*-expressing neurons. Quantification, dendrite length: Student's unpaired t-test (p<0.0001); control (13 neurons, 6 larvae) v. *Rpl22-RNAi* (13 neurons, 5 larvae). Dendrite length of *Rpl22-RNAi*-expressing neurons was quantified using Ppk1::sfGFP because the CD4::tdTomato signal was too

dim to analyze in these mutants. Quantification, dendrite tip number: Mann-Whitney test (p<0.0001); control (13 neurons, 6 larvae) v. *Rpl22-RNAi* (13 neurons, 5 larvae). Quantification, Ppk1::sfGFP signal, dendrites: Student's unpaired t-test (p = 0.8602); control (20 neurons, 9 larvae) v. *Rpl22-RNAi* (27 neurons, 14 larvae). (B) Representative images and quantification of control and *EcR-DN*-expressing neurons. Quantification, dendrite length: Student's unpaired t-test (p<0.0001); control (15 neurons, 9 larvae) v. *EcR-DN* (15 neurons, 7 larvae). Quantification, dendrite tip number: Student's unpaired t-test (p<0.0001); control (15 neurons, 9 larvae) v. *EcR-DN* (15 neurons, 7 larvae). Quantification, Ppk1::sfGFP signal, dendrites: Student's unpaired t-test (p = 0.2318); control (24 neurons, 10 larvae) v. *EcR-DN* (23 neurons, 10 larvae). (C) Representative images and quantification of control neurons and neurons over-expressing *Rac1* (*Rac1* O/E). Quantification, dendrite length: Student's unpaired t-test (p = 0.7618); control (15 neurons, 10 larvae) v. *Rac1* O/E (14 neurons, 5 larvae). Quantification, dendrite tip number: Student's unpaired t-test (p<0.0001); control (15 neurons, 10 larvae) v. *Rac1* O/E (14 neurons, 5 larvae). Quantification, Ppk1::sfGFP signal, dendrites: Student's unpaired t-test (p = 0.0791); control (20 neurons, 9 larvae) v. *Rac1* O/E (20 neurons, 9 larvae). (D) Representative images and quantification of control neurons and neurons over-expressing *PI3K* (*PI3K* O/E). Quantification, dendrite length: Student's unpaired t-test (p = 0.0001); control (12 neurons, 5 larvae) v. *PI3K* O/E (12 neurons, 6 larvae). Quantification, dendrite tip number: Student's unpaired t-test (p = 0.0002); control (12 neurons, 5 larvae) v. *PI3K* O/E (12 neurons, 6 larvae). Quantification, Ppk1::sfGFP (AU), dendrites: Student's unpaired t-test (p<0.0001); control (26 neurons, 14 larvae) v. *PI3K* O/E (22 neurons, 13 larvae). Control genotype: *w1118*; *ppk-Gal4*. Experimental genotypes: *ppk-Gal4* was used to express the indicated construct. *Ppk-CD4::tdTomato* and *Ppk1::sfGFP* included as indicated. Experiments to analyze the effects of *Rpl22-RNAi* and *Rac1 over-expression* were performed together; the controls for these experiments are the same. In the graphs, each data point represents a neuron, and data are plotted as mean ± SEM. n.s. = not significant (p>0.05), ***p = 0.001–0.0001, and ****p<0.0001. AU: arbitrary units. Scale bars, 50 μm.

dendrite growth was normal (Fig 5A) [43]. Although Ppk channels are not essential for dendrite growth, we found, consistent with previous reports, that dendrite arborization was dramatically reduced by the overexpression of a mutant Ppk26 with an amino acid substitution that keeps the channel in an aberrant open state (Ppk26^A547V, also known as the degenerin, or "DEG," mutant due to the degenerative effects of this mutant residue on neuronal morphology; thus, we refer to this Ppk26 transgene as Ppk26^DEG) (Fig 5B) [19,22,44,45]. We attempted to suppress the dendrite arbor phenotype induced by Ppk26^DEG by eliminating Ppk1, which should prevent Ppk26^DEG from reaching the membrane. However, the loss of *ppk1* surprisingly enhanced, rather than suppressed, the Ppk26^DEG dendrite arborization phenotype (Fig 5C). By the 3rd larval instar stage, neurons expressing Ppk26^DEG and lacking Ppk1 had just cell bodies, no dendrites, and also often lacked an axon. A previous study showed that in the absence of Ppk1, Ppk26 that does not reach the membrane accumulates in large, bright puncta [24]. Similarly, we found that fluorescently tagged Ppk26^DEG accumulated in bright puncta in dendrites when *ppk1* was eliminated (Fig 5D; these analyses were carried out in 1st instar larvae, when the mutant neurons still occasionally had dendrites). This suggests that in the absence of Ppk1, the Ppk26^DEG mutant may not reach the dendritic membrane and, thus, may exert its effects on dendrite growth by acting at an internal organelle. Increasing Ppk1 levels partially suppressed the effects of Ppk26^DEG on dendrite growth and fluorescently tagged Ppk26^DEG was normally distributed in dendrites (Fig 5E and 5F). Combined, our results indicate that while Ppk1 and Ppk26 are dispensable for normal dendrite growth, Ppk channel activity must be tightly regulated to achieve proper dendrite arborization. Our results also suggest that mutant, over-active Ppk26^DEG may perturb dendrite growth by acting at internal organelle rather than the cell membrane.

## The transport of Ppk1 from Golgi to dendrites depends on the molecular motor dynein

We next investigated how Ppk1 localizes to dendrites. The majority of transport in neurons occurs along microtubules, which are differentially organized in axons and dendrites. In fly da neurons, nearly all dendritic microtubules are oriented with their minus-ends positioned away from the cell body [46,47]. Thus, the microtubule minus-end-directed motor dynein likely mediates the majority of transport to dendrites [48]. To test whether dynein has a role in transporting Ppk1 to dendrites, we took two approaches to disrupt dynein activity: we used RNAi to reduce the levels of the essential dynein subunit Dynein light intermediate chain (Dlic) and over-expressed dynamitin (Dmn), a dynein co-factor and dynactin complex member;

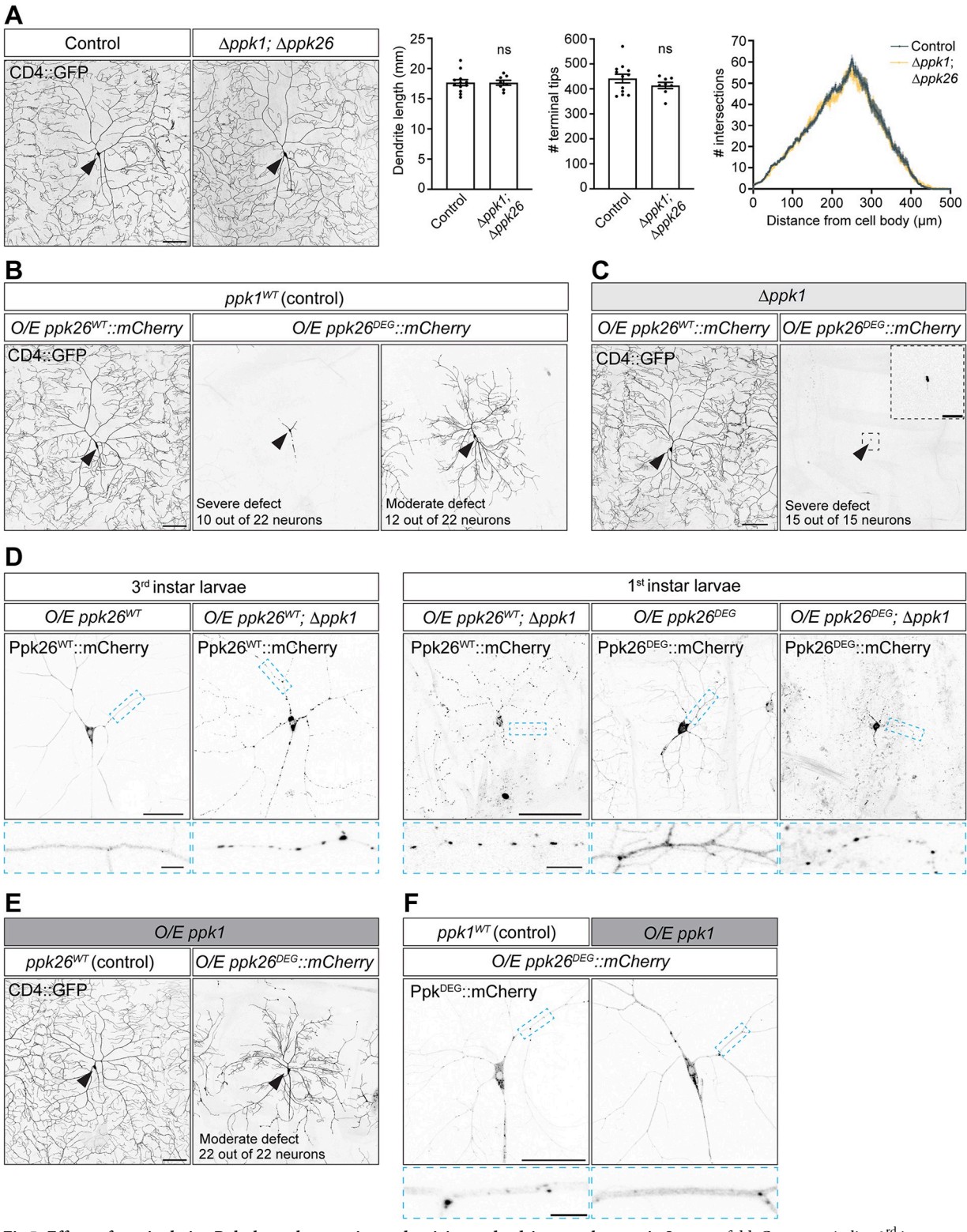

**Fig 5. Effects of manipulating Ppk channel expression and activity on dendrite morphogenesis.** Images of ddaC neurons in live 3rd instar larvae. Neuron morphology is visualized with the neuronal membrane marker CD4::GFP (*ppk-CD4::GFP*) (A-C, E). mCherry-tagged wild-type Ppk26 (Ppk26WT) or Ppk26 with the "degenerin" (DEG) mutation were over-expressed in class IV da neurons as indicated (*ppk-Gal4 UAS-ppk26^WT::mCherry* and *ppk-Gal4 UAS-ppk26^DEG::mCherry*, respectively). (A) Representative images and quantification of control neurons (*w^1118*; 12 neurons, 5 larvae) and neurons lacking both Ppk1 and Ppk26 (*Δppk1; Δppk26: ppk1^attP-KO/attP-KO; ppk26^Δ11/ Δ11*; 8 neurons, 4 larvae).

Quantification, dendrite length: Student's unpaired t-test (p = 0.9783). Quantification, dendrite tip number: Student's unpaired t-test (p = 0.2592). Quantification, Sholl analysis (mean ± SEM): critical radius ($w^{1118}$ = 255 ± 5 μm; $Δppk1$; $Δppk26$ = 249 ± 6 μm; Student's unpaired t-test (p = 0.4550), maximum number of intersections ($w^{1118}$ = 67 ± 2; $Δppk1$; $Δppk26$ = 64 ± 3; Student's unpaired t-test (p = 0.4366). In the dendrite length and dendrite tips graphs, each data point represents a neuron and data are plotted as mean ± SEM. In the Sholl analysis graph, data are plotted as mean ± SEM. n.s. = not significant (p>0.05). (B) Representative images of the morphology of neurons overexpressing Ppk26$^{WT}$ or Ppk26$^{DEG}$. The overexpression of Ppk26$^{DEG}$ results in variable morphologies ranging from severely affected neurons with only short primary dendrites (middle panel) to moderately affected neurons with secondary and higher order dendrites but a reduced dendrite arbor (right panel). (C) Representative images of the morphology of neurons in larvae lacking Ppk1 ($Δppk1$: $ppk1^{attP-KO/attP-KO}$) and, as indicated, overexpressing Ppk26$^{WT}$ or Ppk26$^{DEG}$. In the absence of Ppk1, the overexpression of Ppk26$^{DEG}$ results in a severe phenotype: some ddaC neurons are missing and those that are present have a small cell body with no discernable axon or dendrites. (D) Representative images of the localization of Ppk26$^{WT}$ and Ppk26$^{DEG}$ expressed in control neurons and neurons lacking Ppk1 ($Δppk1$: $ppk1^{attP-KO/attP-KO}$) in 3$^{rd}$ instar (left) and 1$^{st}$ instar (right) larvae. Since neurons lacking Ppk1 and overexpressing Ppk26$^{DEG}$ lack dendritic arbors by 3$^{rd}$ instar (see C), Ppk26 localization was assayed during 1$^{st}$ instar. Loss of Ppk1 results in the accumulation of Ppk26 in puncta. (E) Representative images of the morphology of neurons over-expressing Ppk1 ($ppk-Gal4\ UAS-ppk1$::FLAG) in control neurons and neurons also expressing Ppk26$^{DEG}$. The overexpression of Ppk1 suppresses the severe phenotype generated by Ppk26$^{DEG}$, and all neurons display a moderate phenotype (secondary and higher dendrites are present but the arbor coverage is reduced). (F) Representative images of the localization of Ppk26$^{DEG}$ in a control neuron and neuron over-expressing Ppk1 ($ppk-Gal4\ UAS-ppk1$::FLAG). Scale bars, 100 μm (A, B, C, E), 50 μm (solid-outline panels, D, F), 25 μm (inset, C), and 10 μm (dashed-outline boxes, D, F).

increasing Dmn levels has a dominant-negative effect on dynein activity, likely by perturbing dynein-dynactin interactions [49,50]. We found that Ppk1::sfGFP density was significantly decreased in both *Dlic-RNAi*-expressing neurons and neurons over-expressing *dmn* (Fig 6A). Ppk1::sfGFP also accumulated in axons, although this may be due to the axonal mislocalization of Golgi that occurs when dynein activity is perturbed (disrupting the Golgi-dynein adaptor Lava lamp also resulted in ectopic Golgi outposts in axons and increased axonal Ppk1::sfGFP, which suggests that mis-localization of Golgi alone can affect the axonal density and distribution of Ppk1; S7 Fig). In neurons over-expressing *dmn*, we also observed that younger neurons, which did not display a strong dendrite arborization phenotype, did not have a significant reduction in Ppk1::sfGFP (Fig 6B). This suggests that dendritic Ppk1::sfGFP levels decrease progressively over time as dynein-mediated transport is impeded. Combined, these results indicate that the dendritic localization of Ppk1 and Ppk channels relies on functional dynein.

Given the reduction in dendritic Ppk1::sfGFP in *Dlic-RNAi*-expressing neurons, we next asked where Ppk1 might accumulate when dynein function is disrupted. Most ion channels traffic through the Golgi apparatus, which, in fly da neurons, includes somatic Golgi as well as dendritic Golgi "outposts," which are Golgi mini-stacks found predominantly in the proximal dendritic arbor [51–55]. In control neurons, we observed Ppk1::sfGFP at somatic Golgi but not Golgi outposts; however, in neurons expressing *Dlic-RNAi*, Ppk1::sfGFP accumulated at both somatic Golgi and dendritic Golgi outposts (Fig 6C). These results suggest that Ppk1 traffics through both somatic Golgi and Golgi outposts and depends on dynein to be transported away from Golgi.

We also examined the effects of disrupting dynein activity on the dendritic membrane localization of Ppk1 using Ppk1::GFP(11x3)$^{EC}$. Ppk1::GFP(11x3)$^{EC}$ density increased in neurons expressing *Dlic-RNAi* (Fig 6D). Given that Ppk1::sfGFP density decreased in *Dlic-RNAi*-expressing neurons, we tested whether the GFP(11) tag might affect the stability of Ppk1 in the dendritic membrane. We used an anti-Ppk1 antibody and non-membrane-permeabilizing conditions to probe for membrane-localized Ppk1. As reported by anti-Ppk1, the density of membrane-localized Ppk1 was not affected by the GFP(11) tag (Fig 6E; the anti-Ppk1 signal is absent in the proximal dendrites, cell body, and axon as these are wrapped by glia, which exclude antibodies under the non-membrane-permeabilizing conditions used to probe for Ppk1 [22]). In neurons expressing *Dlic-RNAi*, the dendritic membrane density of Ppk1 read-out by anti-Ppk1 remained equivalent to control levels (Fig 6E). These results suggest that Ppk1::GFP(11x3)$^{EC}$ may be a more sensitive reporter of Ppk1 membrane localization than the

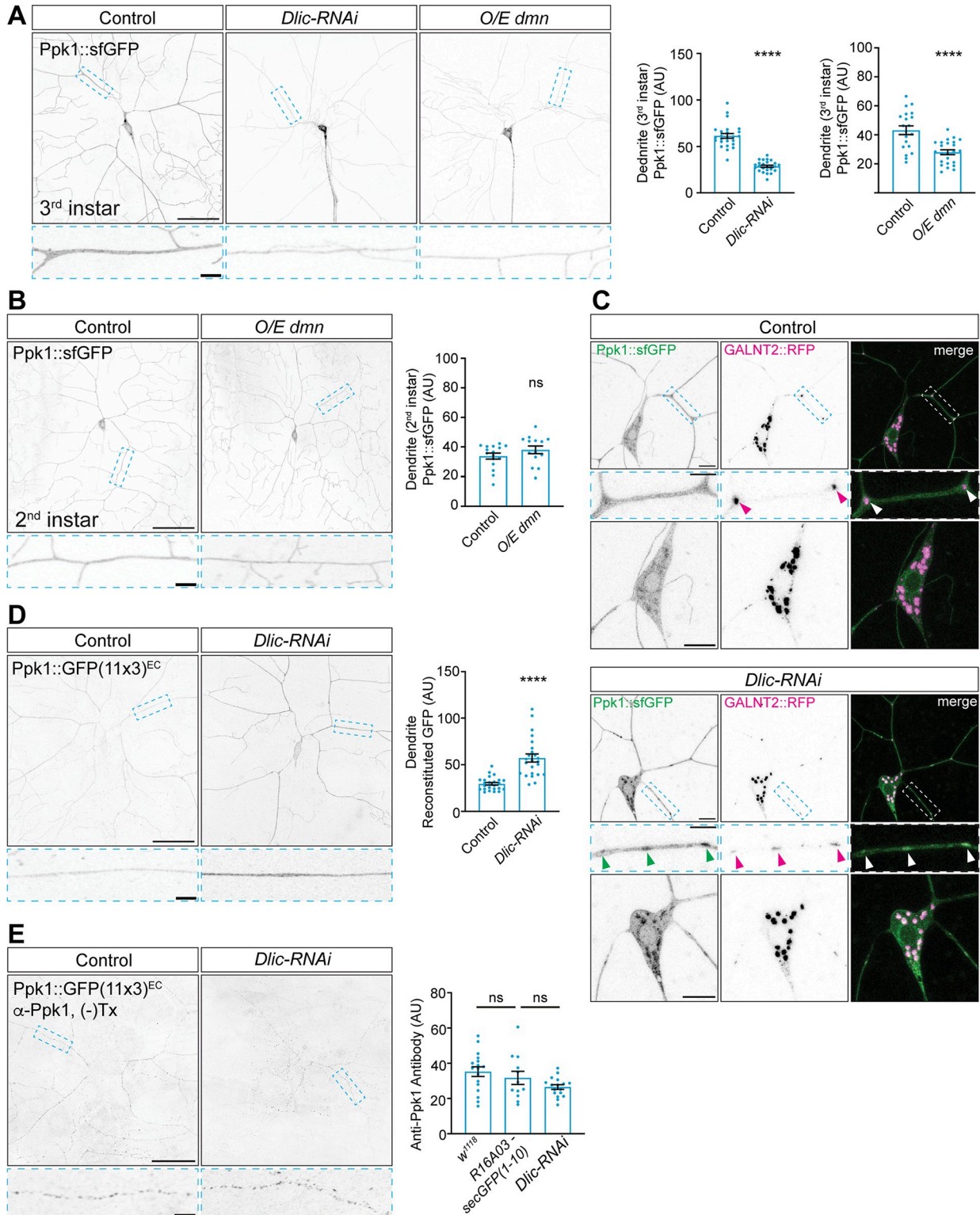

**Fig 6. Ppk1 persists in dendrites when dynein-mediated transport is disrupted, and dynein plays a role transporting Ppk1 away from Golgi.**
Images of ddaC neurons in live 3rd instar (A, C, D) and 2nd instar larvae (B) and fixed 3rd instar larvae (E). SecGFP(1–10) was expressed under the control of an enhancer active in fat bodies (*R16A03-secGFP(1–10)*) (D, E). (A) Representative images and quantification of Ppk1::sfGFP in control neurons and neurons expressing *Dlic-RNAi* or over-expressing *dmn* (O/E *dmn*). Quantification, Ppk1::sfGFP signal: Student's unpaired t-test (p<0.0001), control (21 neurons, 11 larvae) v. *Dlic-RNAi* (21 neurons, 11 larvae); Student's unpaired t-test (p<0.0001); control (20 neurons,

12 larvae) v. O/E *dmn* (23 neurons, 14 larvae). Scale bars, 50 μm (solid-outline panels) and 5 μm (dashed-outline boxes). (B) Representative images and quantification of Ppk1::sfGFP in control neurons and neurons over-expressing *dmn*. Quantification, Ppk1::sfGFP signal: Mann Whitney (p = 0.1417); control (16 neurons, 6 larvae) v. O/E *dmn* (14 neurons, 6 larvae). Scale bars, 50 μm (solid-outline panels) and 5 μm (dashed-outline boxes). (C) Representative images of Ppk1::sfGFP and the Golgi marker GALNT2::TagRFP in control and *Dlic-RNAi*-expressing neurons. In *Dlic-RNAi*-expressing neurons, Ppk1::sfGFP (green arrowheads) accumulates in Golgi in the cell body and Golgi outposts (arrowheads). Scale bars, 10 μm (solid-outline panels) and 5 μm (dashed-outline boxes). (D) Representative images and quantification of reconstituted GFP signal from Ppk1::GFP(11x3)$^{EC}$ in control neurons and neurons expressing *Dlic-RNAi*. Quantification, reconstituted GFP signal: Student's unpaired t-test (p<0.0001); control (24 neurons, 10 larvae) v. *Dlic-RNAi* (24 neurons, 9 larvae). (E) Representative images and quantification of anti-Ppk1 signal in control and *Dlic-RNAi*-expressing neurons. *R16A03-secGFP(1–10)* and *ppk1::GFP(11x3)$^{EC}$::mCherry$^{C-term}$* were included in the R16A03-secGFP(1–10) control and Dlic-RNAi experiment. Quantification, anti-Ppk1 signal: one-way ANOVA with post-hoc Tukey; *w$^{1118}$* control (17 neurons, 6 larvae) v. *R16A03-secGFP(1–10)* control (12 neurons, 4 larvae) (p = 0.617); *R16A03-secGFP(1–10)* control v. *Dlic-RNAi* (17 neurons, 6 larvae) (p = 0.366). Control genotypes: *w$^{1118}$; ppk-Gal4* (A-C), *w$^{1118}$; ppk1::GFP(11x3)$^{EC}$::mCherry$^{C-term}$ R16A03-secGFP(1–10)/+; ppk-Gal4 R16A03-secGFP(1–10)/R16A03-secGFP(1–10)* (D, E), *w$^{1118}$* (E). Experimental genotypes: *w$^{1118}$; ppk-Gal4 UAS-Dlic-RNAi UAS-Dicer* (A, C), *w$^{1118}$; ppk-Gal4 UAS-dmn* (A, B), *w$^{1118}$; ppk1::GFP(11x3)$^{EC}$::mCherry$^{C-term}$ R16A03-secGFP(1–10)/UAS-Dlic-RNAi UAS-Dicer; ppk-Gal4 R16A03-secGFP(1–10)/R16A03-secGFP(1–10)* (D, E). *UAS-GALNT2::TagRFP* and *Ppk1::sfGFP* included as indicated. In the graphs, each data point represents a neuron, and the data are plotted as mean ± SEM. n.s. = not significant (p>0.05) and ****p<0.0001. AU: arbitrary units.

anti-Ppk1 antibody. Our results also suggest that while the overall dendritic levels of Ppk1 decrease when dynein function is perturbed over time (e.g., Ppk1::GFP; Fig 6A), Ppk1 that does make it into the dendritic membrane is not reduced by diminished dynein activity.

## Dendritic density of Ppk1 is reduced by disrupting Rab11

We next asked what transport carriers might be involved in the delivery of Ppk1 to dendrites. We considered Rab11-positive endosomes for several reasons. First, Rab11 has been implicated in the anterograde trafficking of ion channels to the dendritic membrane in mammalian neurons [56], and recent work suggests that Rab11 may affect the trafficking of the Ppk1 partner subunit Ppk26 in fly da neurons [29]. Rab11 also plays a role in the forward trafficking and membrane localization of ENaC family members in the epithelial cells of mammalian kidneys [57–59]. In addition, we, like others, found that disrupting Rab11 function reduces dendrite arborization (Fig 7A) [29,60], which suggests that Rab11 may act in coordinating the dendritic localization of ion channels and dendrite arbor development.

To test whether Rab11 participates in trafficking Ppk1 to dendrites, we perturbed Rab11 function using both *Rab11-RNAi* and a dominant-negative Rab11 construct, Rab11-DN (Rab11-DN carries an S25N mutation that disrupts GTPase activity). Neurons expressing *Rab11-RNAi* or Rab11-DN had decreased dendritic Ppk1 as determined by Ppk1::sfGFP and anti-Ppk1, which was used to quantify Ppk1 membrane density (Fig 7B and 7C) (Ppk1 membrane expression could not be assayed with Ppk1::GFP(11)$^{EC}$ for technical reasons; also, since Rab11-DN is tagged with GFP, we were not able to quantify Ppk1::sfGFP in these neurons) [22,23]. In neurons expressing *Rab11-RNAi*, the decrease in dendritic Ppk1 density was accompanied by an accumulation of Ppk1::sfGFP in the soma (Fig 7C and 7D). The bright puncta of Ppk1::sfGFP in the soma did not overlap with Golgi, indicating that Ppk1::sfGFP was likely accumulating in a post-Golgi vesicle and/or organelle in the *Rab11-RNAi*-expressing neurons (Fig 7D). In addition to the forward trafficking of ion channels, Rab11 is involved in the local recycling of various ion channels in dendrites, typically acting downstream of early endosomes and the GTPase Rab5 [61]. We used a dominant-negative Rab5 to test whether Rab11 might affect Ppk1 membrane density via a local recycling pathway, but overexpression of Rab5-DN did not affect the dendritic membrane density of Ppk1 (Fig 7E). This suggests that the disruption of Rab11 likely reduces the dendritic density of Ppk1 independently of a local recycling pathway; instead, Rab11 may, directly or indirectly, affect the forward trafficking of Ppk1 from the cell body to dendrites.

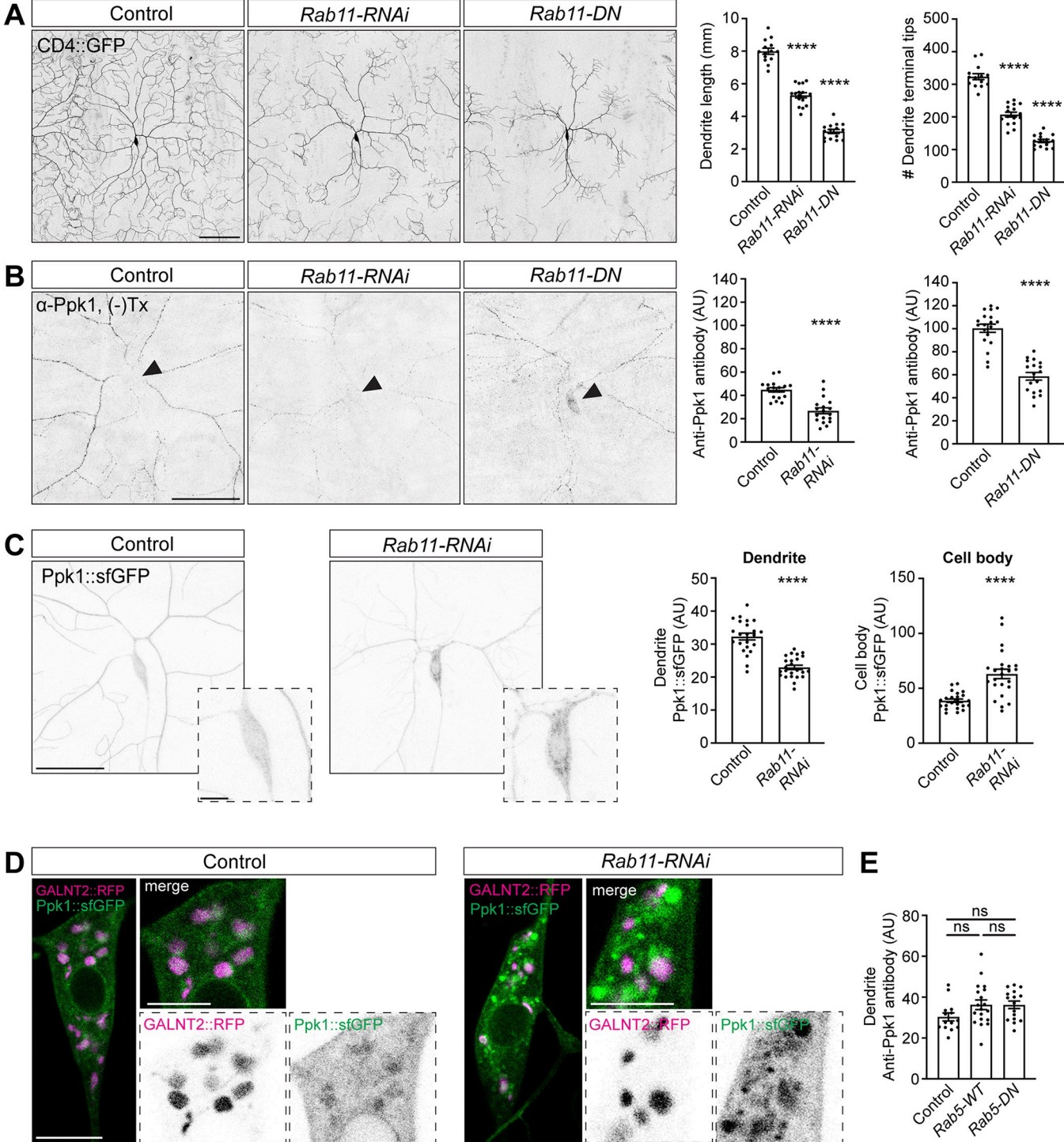

**Fig 7. Disrupting Rab11 reduces dendrite arbor growth and alters Ppk1::GFP localization.** (A) Representative images and quantification of dendrite length and dendrite tip number in control neurons (14 neurons, 6 larvae) and neurons expressing *Rab11-RNAi* (16 neurons, 7 larvae) or *Rab11-DN* (15 neurons, 7 larvae) in live 2$^{nd}$ instar (72 h AEL) larvae. Quantification, dendrite length: One-way ANOVA with post-hoc Tukey; control v. *Rab11-RNAi* (p<0.0001), control v. *Rab11-DN* (p<0.0001). Quantification, dendrite tips: One-way ANOVA with post-hoc Tukey: control v. *Rab11-RNAi* (p<0.0001), control v. *Rab11-DN* (p<0.0001). (B) Representative images and quantification of anti-Ppk1 signal in control neurons and neurons expressing *Rab11-RNAi* or *Rab11-DN* in 3$^{rd}$ instar larvae fixed and immunostained under non-permeabilizing conditions ([–]Tx). Under these non-permeabilizing conditions, the anti-Ppk1 antibody is unable to access the glial-ensheathed proximal dendrites, cell body, and axon. Quantification, anti-Ppk1 signal, dendrite: Student's unpaired t-test (p<0.0001); control (18 neurons, 6 larvae) v. *Rab11-RNAi* (18 neurons, 6 larvae), and Student's unpaired t-test (p<0.0001); control (19 neurons, 6 larvae) v. *Rab11-DN* (18 neurons, 6 larvae). Different aliquots of the anti-Ppk1 antibody were used in the experiments with *Rab11-RNAi* and *Rab11-DN*, resulting in

different baseline anti-Ppk1 signal intensities. In neurons expressing *Rab11-DN*, anti-Ppk1 signal is often seen in the cell body (arrowhead). (C) Representative images and quantification of Ppk1::sfGFP in the dendrites and cell bodies of control (23 neurons, 18 larvae) and *Rab11-RNAi*-expressing neurons (24 neurons, 12 larvae) in live 3$^{rd}$ instar larvae. Quantification, Ppk1::sfGFP, dendrite (left) and cell body (right): Student's unpaired t-test (p<0.0001) for both. (D) Representative images Ppk1::sfGFP and Golgi marked by GALNT2::TagRFP in the cell bodies of control and *Rab11 RNAi*-expressing neurons in live 3$^{rd}$ instar larvae. (E) Quantification of anti-Ppk1 signal in control neurons (14 neurons, 6 larvae) and neurons over-expressing wild-type Rab (Rab-WT; 19 neurons, 6 larvae) or a dominant-negative Rab5 mutant (Rab-DN; 16 neurons, 6 larvae). Quantification: one-way ANOVA with post-hoc Tukey; control v. *Rab5-WT* (p = 0.1323), control v. *Rab5-DN* (p = 0.1562), and *Rab5-WT* v. *Rab5-DN* (p = 0.9999). Control genotype: *w$^{1118}$; ppk-Gal4* (A-E). Experimental genotypes: *w$^{1118}$; ppk-Gal4 UAS-Rab11-RNAi* (A-D), *w$^{1118}$; ppk-Gal4 UAS-Rab11-DN::GFP* (B), *w$^{1118}$; ppk-Gal4 UAS-Rab5-WT::GFP* (E), and *w$^{1118}$; ppk-Gal4 UAS-Rab5-DN::GFP* (E). *Ppk1::sfGFP* included as indicated. (-)Tx: no Triton X-100, indicates non-permeabilizing conditions. In the graphs, each data point represents a neuron, and data are plotted as mean ± SEM. AU: arbitrary units. n.s. = not significant (p>0.05) and ****p<0.0001. Scale bars, 50 µm (solid-outline panels, A-C), 10 µm (dashed-outline boxes in C and panels in D), and 5 µm (dashed-outline boxes in D).

## Discussion

The perception of stimuli by sensory neurons depends on the morphogenesis of a dendritic arbor equipped with ion channels and receptors that will detect sensory inputs. Central outstanding questions include: How are ion channels localized to sensory neuron dendrites during development, and how does a sensory neuron properly match ion channel density with dendrite arbor size? Sensory dendrites lack synaptic input, which has made it unclear whether known mechanisms of ion channel trafficking to dendrites with synapses would also regulate the delivery of ion channels to the asynaptic dendrites of sensory neurons. Our studies of the localization of endogenous Ppk1 suggest a model in which sensory neurons may package some ion channels into the membrane that drives dendritic arbor expansion, thus coordinating the delivery of ion channels with arbor growth to ensure proper channel density.

We monitored the localization of endogenous Ppk1 using several different fluorescent protein tags, including a novel secreted split-GFP approach that illuminates when Ppk1 is inserted into the neuronal cell membrane. Tagging endogenous Ppk1 revealed that it is present in dendrites, as previously reported, but, unexpectedly, also axons and axon terminals. Our experiments with Ppk1::GFP(11)$^{EC}$ indicate that Ppk1 is present throughout the somatodendritic membrane but not the membrane of the proximal axon or axon shaft. The clear demarcation of Ppk1::GFP(11)$^{EC}$ fluorescent signal between the somatodendritic and axonal compartments is consistent with the presence of a diffusion barrier between these two compartments in the proximal axon of fly neurons, as was previously indicated by studies of the fly ankyrin Ank2 [62–64]. Although there is a barrier to the diffusion of Ppk1 from the somatodendritic membrane to the axonal membrane, Ppk1 is not prevented from being trafficked into the axon to the axon terminal. The sensory neuron axon terminals are located in the ventral nerve cord, which is ensheathed by glia [65–67]. Given that these glia likely exclude secGFP(1–10), the Ppk1::GFP(11)$^{EC}$ fluorescent signal at axon terminals may indicate that Ppk1 is trans-endocytosed from dendrites to axons. It is not clear what role (if any) Ppk1 may play in the axon terminals of class IV da neurons, although other studies, including in flies, support presynaptic roles for Ppk channels and their orthologs [19,20,68]. Additional work will be needed to determine whether Ppk channels at axon terminals contribute to Ppk channel function in da neurons.

Our data reveal that Ppk1 is present throughout the membrane of dendrites as they grow, indicating that Ppk channels are an integral component of both newly formed and extending dendrites. While the localization and function of ion channels in growing axons is well established [69–75], little is still known about the localization and function of ion channels in developing dendrites. The timing and breadth of Ppk1 distribution indicates that the class IV da sensory dendrites are likely equipped with the capacity to detect stimuli as soon as dendrites emerge. Indeed, recent work has implicated members of the *C. elegans* DEG/ENaC family in sensing mechanical forces to promote terminal branch growth during arbor formation in

PVD neurons [76]. Although dendrite development occurred normally without Ppk1 and Ppk26, it is nonetheless possible that Ppk channels participate in dendrite growth, possibly in collaboration with another (mechanosensory) ion channel. In vertebrate neurons, the distribution of ion channels in young dendrites has been reported but is not well characterized, although a rich body of work supports the role of channel activity in regulating dendrite growth [77–79]. Our visualization of fluorescently tagged endogenous Ppk1 provides evidence that ion channels are part of the growing dendritic membrane, similar to the localization of ion channels in axons, and that sensory dendrites thus have a "built in" capacity to detect stimuli.

The presence of Ppk1 in growing dendrites raises the possibility that Ppk channels are localized to dendrites as an integral component of the membrane that spurs expansive arbor growth. Recent qRT-PCR analysis revealed that Ppk1 mRNA levels are high during the period of intense dendrite growth that establishes dendrite arbor size but *ppk1* mRNA levels drastically decrease when dendrite growth slows and switches to a proportional growth phase that maintains arbor size [6,27]. The molecular motor dynein, which is the predominant motor for transport for dendrites, is needed for both the growth and maintenance of dendrites [48,80], and the reduction of Ppk1::sfGFP in neurons expressing *Dlic-RNAi* or over-expressing *dmn* indicates that dynein has a role in transporting Ppk1 (and Ppk channels) to dendrites (S8 Fig). Our experiments also suggest that the effects of disrupting dynein are progressive. When neurons overexpressing *dmn* are young and still retain some dynein activity, Ppk1 is initially transported to dendrites, where Ppk1 remains stably integrated in the dendritic membrane even as dynein activity diminishes over time. The idea that Ppk channels are stably integrated into the dendritic membrane with little turnover is supported by our finding that Ppk1 density is not significantly affected when local membrane protein recycling is disrupted via the expression of Rab5-DN. Our data are consistent with the model that Ppk1 is stably integrated into the dendritic membrane as dendrites grow expansively, and, once dendrite growth plateaus, there is little replenishment.

Our data indicate that Rab11 plays a role in the transport of Ppk1 to dendrites, either directly or indirectly. Rab11 is an integral component of recycling endosomes and has also been implicated in the anterograde trafficking of receptors and ion channels, including ENaCs, in neuronal and non-neuronal cells [56,57,81–83]. Our results implicate Rab11 in the forward transport of Ppk1 in fly da neurons, which points to the potential conservation of DEG/ENaC/ASIC trafficking pathways across organisms and cell types. Disrupting Rab11 does not lead to a total loss of Ppk1 from dendrites, which is similar to our findings when we disrupt dynein activity. This may be due to incomplete perturbation of Rab11 or the function of Rab11-positive endosomes, or it could indicate a complementary pathway for the transport of Ppk1 and Ppk channels to dendrites. The pathways that supply membrane and membrane proteins to dendrites, particularly growing dendrites, are still poorly understood. Our studies suggest that in sensory neurons, expansive dendrite growth is coordinated with the delivery of ion channels, such as Ppk, thus ensuring a proper density of ion channels for normal neuronal function.

## Materials and methods

### Fly husbandry and stocks

Fruit flies were maintained at 25°C on cornmeal-molasses-yeast medium. The generation of new *ppk1* alleles, *UAS-secGFP(1–10)*, and *R16A03-secGFP(1–10)* flies are described below. *ppk26* strains including *ppk26^{Δ11}*, *UAS-ppk26::mCherry*, and *UAS-ppk26-DEG(A547V)::mCherry* were gifts of Dr. Yuh Nung Jan (UCSF) [22]. The following alleles and transgenic fly

strains were obtained from the Bloomington Drosophila Stock Center (BDSC), Vienna Dro-sophila Resource Center (VDRC), and individual laboratories: *ppk-Cas9* [36], *ppk-CD4::tdTo-mato* (BDSC 35845), *ppk-GFP(11)^EC^::CD4::tdTomato* [35], *hsp70-Cre* (BDSC 1092), *DcG-Gal4* [84], *UAS-Dcr-2* (BDSC 24650), *UAS-Dlic-RNAi* (VDRC 41686), *UAS-dynamitin* (BDSC 8784), *UAS-EcR-DN* (BDSC 9449), *UAS-GALNT2::TagRFP* (BDSC 65253), *UAS-sfGFP(1–10)* (Bo Huang, UCSF), *UAS-lva-DN* (BDSC 55055), *Ppk-Gal4* (BDSC 32078, BDSC 32079), *UASp-Rab5-WT::YFP* (BDSC 24616), *UASp-Rab5-DN[S43N]::YFP* (BDSC 9772), *UAS-GFP::Rab5* (BDSC 43336), *UAS-Rab7::GFP* (BDSC 42705), *UAS-ppk1::FLAG* (BDSC 79619), *UAS-PI3K* (BDSC 8294), *UAS-Rab11-RNAi* (VDRC 108382), *UAS-Rab11-DN(3–4)::GFP* (Hsiu-Hsiang Lee, National Taiwan University College of Medicine), *UAS-Rac1* (BDSC 6293), *UAS-RpL22 RNAi* (BDSC 34828), *U6-Sec23-gRNA* (BDSC 79400), *GFP::TrpA1* (BDSC 61795), *w^1118^*.

## Generation of *ppk1* alleles

The endogenous *ppk1* gene (~3.7 kb encompassing the entire *ppk1* transcript) was knocked-out and replaced with an attP site to facilitate the reliable integration of new *ppk1* alleles. The *ppk1^attP-KO^* strain was generated using CRISPR-Cas9 genome engineering and ends-out gene-targeting [85–87]. We used two guide RNAs that flank *ppk1* (upstream gRNA: 5'-GTTCTTA-TATCTAGAGATGT-3', and downstream gRNA: 5'-GTCAAGACTTGAAGAATACTG-3') and a donor template, which contained homology arms surrounding an attP site and 3xP3-DsRed flanked by loxP sites. Candidate strains were identified by DsRed expression in adult eyes, and the *ppk1* locus was analyzed by sequencing genomic DNA from isogenized candidate strains. 3xP3-DsRed was then removed by crossing to flies expressing Cre recombinase. A single strain with the desired replacement of *ppk1* with an attP site was used to generate all knock-in alleles.

Constructs to create knock-in alleles were generated using standard molecular biology techniques and Gibson Assembly to add molecular tags. Two plasmid backbones were used: *pGE-attB-GMR* [86], which includes a *GMR-mini-white^+^* cassette to identify knock-in alleles by red eye color in adults, and *pBSK-attB-3xP3* (this study), which includes a *3xP3-DsRed* cassette to identify knock-in alleles by DsRed expression in adult eyes. *pBSK-attB-3xP3* was generated by adding an *attB* site and *3xP3-DsRed* to *pBSK*. All the exogenous sequences (e.g., the product of attB/attP recombination) knocked-into the endogenous *ppk1* locus were the same regardless of which plasmid backbone was used. New *ppk1* alleles in the *pGE-attB-GMR* vector were first subcloned into *pBSK*, modified, and then inserted into *pGE-attB-GMR* using EcoRI and KpnI. New *ppk1* alleles in the *pBSK-attB-3xP3* plasmid were cloned directly using Gibson Assembly (*pBSK-attB-3xP3-ppk^WT^* was the starting plasmid for many of the knock-in alleles). All constructs were verified by sequencing prior to injection. *attB*-containing plasmids with *ppk1* knock-in alleles were injected into *ppk1^attP-KO^* embryos expressing ΦC31 integrase (BestGene Inc., Chino Hills, CA). The *GMR-mini-w^+^* and *3xP3-DsRed* markers were subsequently removed by crossing to flies expressing Cre recombinase.

The following alleles were generated in this study: First a wild-type knock-in allele (*ppk1^WT-k'in^*) was generated by cloning the part of the *ppk1* locus that was eliminated by the replacement strategy (plasmid *pGE-attB-GMR-ppk1^WT^*). The resulting *ppk1^WT-K'in^* flies displayed no overt phenotypes and restored the normal pattern of *ppk1* expression. sfGFP::Ppk1 and Ppk1::sfGFP were created by adding one copy of sfGFP and a GGS(x4) linker at the N- or C-terminus, respectively, of Ppk1 (plasmids *pGE-attB-GMR-sfGFP::ppk1* and *pGE-attB-GMR-ppk1::sfGFP*). A similar approach was used to generate Ppk1::mCherry and sfGFP::Ppk1::mCherry (plasmids *pBSK-attB-3xP3-ppk1::mCherry* and *pBSK-attB-3xP3-sfGFP::ppk1::mCherry*). Ppk1::sfGFP^EC^-Site 1 and Ppk1::sfGFP^EC^-Site 2 were created by tagging Ppk1 with one copy of sfGFP at either

extracellular Site 1 (between Asn171 and Ile172) or Site 2 (between Gln204 and Leu205). sfGFP was flanked on both sides by a GGS(x4) linker (plasmids *pBSK-attB-3xP3-ppk1::sfGFP^EC-Site 1* and *pBSK-attB-3xP3-ppk1::sfGFP^EC-Site 2*). Ppk1::GFP(11x3)^EC was created by tagging Ppk1 with three copies of the split-GFP peptide GFP(11) at extracellular Site 1 (between Asn171 and Ile172). The three copies of GFP(11) were flanked on both sides by a GGS(x4) linker (plasmid *pBSK-attB-3xP3-ppk1::GFP(11x3)^EC*). Ppk1::GFP(11x3)^EC::mCherry^C-term was created by tagging Ppk1::GFP(11x3)^EC at the C-terminus with one copy of mCherry connected by a GGS(x4) linker (plasmid *pBSK-attB-3xP3-ppk1::GFP(11x3)^EC::mCherry^C-term*). Ppk1::pHluorin^EC::mScarlet^C-term and Ppk1::pHluorin^EC::mCherry^C-term were created by adding one copy of supercleptic pHluorin (synthesized as a gene block by GeneWiz, South Plainfield, NJ) at extracellular Site 1 (between Asn171 and Ile172) and flanked on both sides by a GGS(x4) linker and tagged at the C-terminus with either one copy of mScarlet-I (synthetic gene block from GeneWiz) or one copy of mCherry connected with a GGS(x4) linker (plasmids *pGE-attB-GMR-ppk1::pHluorin^EC::mScarlet^C-term* and *pBSK-attB-3xP3-ppk1::pHluorin^EC::mCherry^C-term*).

## Generation of the *UAS-secGFP(1–10)* and *R16A03-secGFP(1–10)* transgenic fly strains

*UAS-secGFP(1–10)*: The GFP(1–10) coding sequence, which was synthesized as a gBlock fragment (Integrated DNA Technologies, Inc.), was PCR-amplified and cloned into the NheI/XbaI sites of *pIHEU-sfGFP-LactC1C2* [88]. The resulting *pIHEU-secGFP(1–10)* construct contains a signal-peptide sequence from Adipokinetic hormone fused in-frame before GFP(1–10). The construct was integrated at the *attP^VK00005* site in the Drosophila genome (injected by Rainbow Transgenic Flies, Inc).

*R16A03-secGFP(1–10)*: In a synthesis-based approach, the UAS sequence in *pIHEU-secGFP(1–10)* was replaced with the *R16A03* enhancer, which is active in fat bodies (Genewiz). The resulting *pIHEU-R16A03-secGFP(1–10)* construct was integrated at the *attP40* and *attP^VK00027* sites in the Drosophila genome (injected by BestGene, Inc).

## Fixation and Immunohistochemistry

To visualize Ppk channel expression in larval fillets, wandering third instar larvae were washed in 1X PBS (phosphate buffered saline, pH 7.4), dissected in PHEM buffer (80 mM PIPES pH 6.9, 25 mM HEPES pH 7.0, 7 mM $MgCl_2$, 1 mM EGTA) and fixed in 4% paraformaldehyde in 1X PBS with 3.2% sucrose for 20 minutes. For the dissection, larvae were pinned onto a Sylgard plate with their dorsal trachea facing down and were cut on their ventral side to preserve the ddaC neurons. After fixation, the dissected fillets were washed 3 times with 1X PBS, quenched with 50 mM $NH_4Cl$ for 10 minutes, and blocked in blocking buffer composed of 2.5% bovine serum albumin (BSA; catalog number A9647, Sigma), 0.25% fish-skin gelatin (FSG; catalog number G7765, Sigma), 10 mM glycine, and 50 mM $NH_4Cl$ for 3 hours at room temperature. Fillets were incubated in primary antibody diluted in blocking buffer overnight at 4˚C. The next day, fillets were washed in 1X PBS at room temperature (3 x 30 minutes) and incubated with secondary antibody diluted in blocking buffer overnight at 4˚C. The next day, fillets were washed in 1X PBS at room temperature (3 x 30 minutes) and mounted onto glass microscope slides (Fisher Scientific, Selectfrost, 25x75x1.0 mm) with cover glass (Fisher Scientific 24x50-1.5) using elvanol containing antifade (polyvinyl alcohol, Tris-HCl pH 8.5, glycerol and DABCO, catalog number 11247100, Fisher Scientific, Hampton, NH). All wash and incubation steps were performed on a nutator. To visualize membrane-localized Ppk1, a rabbit anti-Ppk1 antibody (1:3000; gift of Yuh Nung Jan, UCSF) [22] targeting an extracellular epitope of Ppk1

was used without detergent in any wash or incubation steps. A fluorescently conjugated secondary antibody was used: goat anti-rabbit-DyLight 633 (1:500; catalog #35563, Invitrogen).

To visualize Ppk expression in the ventral nerve cord, brains from wandering third instar larvae were isolated from larval carcasses in 1x PBS. Following fixation (4% paraformaldehyde in 1X PBS with 3.2% sucrose for 15 minutes), brains were washed (3 X 5 minutes) in 1x PBS and mounted with the optic lobes facing down. The cover glass was stabilized with four small dots of vacuum grease spacers in four corners of the slide.

To visualize Ppk1::sfGFP in young ddaC neurons, embryos were collected on grape plates for several hours and then devitalized in a solution of 50% bleach and 50% $H_2O$ for 2–3 minutes. The eggshells were washed away by rinsing with $H_2O$ and then placed in a tube containing equal quantities of n-heptane and fixative (4% paraformaldehyde in 1X PBS) for 10 minutes. Fixed embryos were washed with 1X PBS with 0.1% Triton X-100 (3 x 10 minutes) and then probed for 2 hours with goat anti-HRP conjugated Alexa Fluor 647 (1:1000, or 0.5 mg/mL, Jackson ImmunoResearch, West Grove, PA). The anti-HRP antibody recognizes a glycoprotein epitope that is present throughout the fruit fly nervous system, enabling visualization of virtually all neuronal membranes. Following incubation with the anti-HRP antibody, the embryos were washed in 1X PBS with 0.1% Triton X-100 (3 x 30 minutes). Embryos were mounted in a solution of 50% glycerol and 50% 1X PBS on glass microscope slides (Fisher Scientific, Selectfrost, 25x75x1.0 mm) with cover glass (Global Scientific, 24x50 mm-1.5). All steps were performed at room temperature.

## Imaging

Imaging was performed on either a SP5 or Stellaris laser-scanning confocal microscope (Leica Microsystems) with sensitive hybrid (HyD) and photomultiplier tube (PMT) detectors using 20×0.7 NA (SP5), 20×0.75 NA (Stellaris), and 40×1.3 NA (SP5 and Stellaris) oil-immersion objectives. The dorsal class IV da neurons (ddaCs) in abdominal segments A2-A5 of control and mutant larvae were imaged. For live imaging, individual larvae were placed into a small drop of 50% glycerol:1X PBS solution that was flanked on both sides by strips of vacuum grease spacers. The larva was then immobilized by pressing a cover glass on top of the spacers. The larva was oriented with its dorsal trachea facing up and rolled gently to one side for optimal positioning of the ddaC neurons. Fixed samples (larval fillets, VNCs) were imaged using a 40×1.3 NA oil-immersion objective. Images were collected via z-stacks (1024x1024-pixel resolution, 1 μm per z-step). Movies of Ppk::GFP(11)EC in growing dendrite tips were collected in 72 h AEL larvae using a 40×1.3 NA oil-immersion objective at a resolution of 1024 x 256 pixels, zoom 6, and a rate of 0.34 frames per second (2.942 seconds per frame) for a duration of 3 minutes. For FRAP: First, a pre-bleach z-stack was obtained of the dendrite region to be bleached, which was a secondary dendrite segment longer than 50 μm without branch points, visible in a single z-plane, and within 150 μm of the cell body. A 50 μm circular region of interest (ROI) was centered on the dendrite segment. Next, the Leica FRAP Wizard was used to bleach the ROI: pre-bleach (10 frames; 0.739 sec/frame), bleach at 100% 488 laser intensity (10 frames; 0.739 sec/frame), post-bleach (10 frames; 0.739 sec/frame). After bleaching, z-stacks (z-step size of 0.5 μm) were captured at 1, 3, 5, 10, and 20 minutes post-bleaching. For all experiments, the same imaging settings were used for control and experimental conditions. Images and movies were subsequently analyzed using FIJI or Metamorph.

## Quantification of Ppk1 signal

Ppk1 levels were measured in FIJI using the following reporters: anti-Ppk1 antibody, Ppk1::sfGFP, and Ppk1::sfGFP(11x3)EC::mCherryC-term. First, maximum intensity projections of z-

stack images were generated. To quantify Ppk1 in dendrites, the fluorescence intensity of three different dendrite branches was quantified by tracing 50-μm lines over segments close to the cell body and averaging the signal intensity along each segment. The average intensity of a 50-μm line traced over the background was subtracted from each dendrite trace. Under non-permeabilizing conditions, anti-Ppk1 signal was absent in the proximal dendrites (likely due to ensheathing glia preventing antibody access), and therefore dendrite traces initiated ~10–15 μm away from the cell body. An average intensity for each neuron was quantified by averaging the intensities of the three dendrite segments after subtracting the background signal. A similar protocol was used to determine the fluorescence intensity of Ppk1::sfGFP in axons: a 50-μm line was traced over the axon close to the cell body and the average intensity of a 50-μm line traced over the background was subtracted.

To measure the extent of Ppk1::sfGFP signal in growing dendrites over time, z-stack images of Ppk1::sfGFP and CD4::tdTomato taken at two time points 30 sec apart were aligned using the bUnwarpJ plugin to generate a composite image representing the change in dendrite length and fluorescent signal over time. Dendrite length was quantified based on the CD4::tdTomato signal, and the percentage of dendrite that was Ppk1::sfGFP-positive was calculated.

## Quantification of dendrite morphology

Imaris software with Filament Tracer (version 9.7–9.8, Oxford Instruments) was used to quantify dendrite length and the number of terminal tips. Neurons were analyzed in larvae that were aged to 72 h AEL unless otherwise mentioned. To capture the entire ddaC dendritic arbor, z-stacks (1024x1024-pixel resolution, 1 μm per z-step) of neurons expressing fluorescent membrane markers were captured using a 20×0.7 NA (Leica SP5) 20×0.75 NA (Leica Stellaris) oil-immersion objectives. Maximum intensity projections of the z-stack images were created in FIJI, and neighboring neurons were cropped out using the freehand draw tool. These images were then further processed in FIJI by applying a threshold to eliminate background signal. The images were imported into Imaris, and Filament Tracer (BitPlane) with automatic detection was used to quantify total dendrite length and the number of terminal tips. The largest and smallest diameters of each neuron were manually measured to generate the dendrite start points and seed points. The thresholds were manually adjusted for the start points and seed points in order to cover the entire arbor and to reduce background points; seed points were manually added to segments that were not automatically identified. The filament was edited to remove the axon segment and to correct misdrawn segments. Measurements generated in Imaris were exported to Excel for further analysis. Sholl analysis of dendritic arbors, which quantifies the number of dendrites in an arbor by counting branches that intersect with concentric circles centered on the cell body, was also carried out in Imaris. The critical radius is the circle with the maximum number of dendritic branch intersections and the number of intersections at this radius are reported as the maximum number of branches.

## Quantification of FRAP

Analysis was performed in FIJI by creating maximum projections of the z-stacks from each time point. A line trace through the bleached region was drawn and the average intensity (arbitrary units; AU) of the center 10 μm was used to quantify the signal recovery over time. To account for general photobleaching, the signal intensity in the bleached 10 μm section was normalized by dividing by the average intensity (AU) of a 10 μm segment in a different secondary branch outside of the bleached region. Average intensity values were exported to Excel for further analysis. To calculate percent recovery of signal after photobleaching, the normalized average signal of the 10 μm branched region at 1, 3, 5, 10, and 20 minutes was divided by the

initial signal from the pre-bleach z-stack. The numerical data for these experiments are included in S1 Table.

## Statistical analysis

All data were blinded prior to analysis. Statistical analysis was performed in Excel and Graph-Pad Prism using a significance level of $p < 0.05$. Outliers were identified using Grubbs' test and removed. Data were analyzed for normality using the Shapiro-Wilk test. Normally distributed data were then analyzed for equal variance and significance using either an F-test and Student's unpaired t-test (two samples) or one-way ANOVA with post-hoc Tukey (multiple samples). Data sets that were not normally distributed were analyzed using Mann-Whitney U test (two samples) or Kruskal-Wallis test with post-hoc Dunn test for significance (multiple samples). Significance levels are represented as follows: not significant (ns), $p > 0.05$; *, $p = 0.05–0.01$; **, $p = 0.01–0.001$; ***, $p = 0.001–0.0001$; and ****, $p < 0.0001$. Data are presented as the mean ± standard error of the mean (SEM) unless otherwise noted. In the graphs, *n* represents a neuron unless otherwise indicated.

## Supporting information

**S1 Fig. Effects of tagging endogenous Ppk1 on Ppk1 levels.** Representative images and quantification of membrane-expressed Ppk1, recognized by anti Ppk1 antibodies (top) and sfGFP-tagged Ppk1 (bottom) in control neurons *(w1118;12* larvae, 36 neurons) and neurons heterozygous for Ppk1 tagged at the **N-** or C-terminus (sfGFP::Ppk1 and Ppk1:: sfGFP, respectively) *(sfGFP::ppk1;* 11 larvae, 33 neurons and *ppk1:: sfGFP;* 11 larvae, 33 neurons). Quantification, Ppk1 membrane levels (top graph): One-way ANOVA with post-hoc Tukey: control v. *sfGFP::ppk1* (p = 0.2570), control v. *ppk1::sfGFP* (p = 0.9097), *ppk1::sfGFP* v. *sfGFP::ppk1* (p = 0.4802). Quantification, sfGFP-tagged Ppk1 (bottom graph): Student's unpaired t-test (p = 0.1707). In the graphs, each data point represents the average signal intensity per larva (2–3 neurons per larva). Data are plotted as mean± SEM. n.s. = not significant (p>0.05). AU: arbitrary units. Scale bar, 50 μm.
(PDF)

**S2 Fig. Localization of Ppk1 tagged with mCherry.** Representative images of ddaC neurons in live 3rd instar larvae (A, B, D-E) and axon terminals in fixed ventral nerve cords (VNCs) (C). Dashed-outline boxes: zoomed-in views of dendrite branch and cell bodies. (A) Representative image of Ppk1::mCherry. Dashed-outline boxes: Individual 1-μm thick z-plane zoomed in views of dendrites and axons; a line indicates the position at which an intensity profile plot was generated. Scale bars, 50 μm and 5 μm (dashedoutline boxes). (B) Representative images of the cell body and proximal dendrites of Ppk1::sfGFP, Ppk1::mCherry, and sfGFP::Ppk1:: mCherry. Scale bar, 10 μm. (C) Representative images of dual-tagged sfGFP::Ppk1::mCherry in dendrites, cell body, proximal axon, and axon terminals in the VNC. Scale bars, 50 μm (left), 5 μm (middle dashed-outline boxes), 10 μm (right). (D) Representative images of Ppk1:: mCherry relative to Rab5::GFP (top) and Rab7::GFP (bottom). Arrowheads: Colocalized signal. Scale bars, 10 μm. (E) Representative images and quantification of Ppk1::mCherry in control (20 neurons, 10 larvae) and Rab5-DN-expressing neurons (20 neurons, 10 larvae). Quantification, Ppk1::mCherry puncta number: Mann-Whitney test (p<0.0001). Ppk1:: mCherry puncta were quantified in dendrites within 70 μm of the cell body. Control genotype: *w1118*; *ppk-Gal4*. Experimental genotype: *w1118*; *ppk-Gal4 UAS-Rab5-DN::YFP* (E). *UAS-Rab5*::*GFP, UAS-Rab7::GFP*, and *ppk1::mCherry* included as indicated (D-E). Scale bar 10 μm. In the graphs, each data point represents a neuron, and data are plotted as mean ±SEM.

****p<0.0001. AU: arbitrary units.
(PDF)

**S3 Fig. Endogenous Ppk1 tagged with sfGFP and pHluorin at extracellular positions.** (A) Representative images of ddaC neurons expressing Ppk1 tagged with one copy of sfGFP at two different extracellular (EC) sites, Site 1 (between Asn171 and Ile172) and Site 2 (between Gln204 and Leu205). Cartoon (left) shows the crystal structure of an individual cASIC1 subunit (PDB: 2QTS)[18]. Ppk1 is predicted to have a similar structure); three subunits compose a channel. The amino acid sequence for *D. melanogaster* is superimposed on the cASIC cartoon structure, and the sites that correspond to where sfGFP was inserted into Ppk1 are indicated. The locations of Site 1 and Site 2 were predicted by aligning the amino acid sequences of Ppk1 and cASIC1. Ppk1 tagged with sfGFP at Site 1 showed similar fluorescent signal as Ppk1 tagged with sfGFP at the N- or Cterminus; thus, Site 1 was used for the insertion of additional tags (e.g., superecliptic pHluorin and GFP(11)). Site 2 is located near the position at which a haemagglutinin (HA) tag was inserted in rASIC1a [32]. (B) Representative image of a ddaC neuron expressing Ppk1 tagged with one copy of pHluorin at Site 1 and mCherry at the C-terminus (Ppk1:: pHluorinEC::mCherryC-term). The neuron is heterozygous for ppk1:: pHluorinEC::mCherryC-term. Scale bar, 50 μm. (C) Representative images of ddaC neurons expressing Ppk1 tagged with one copy of pHluorin at Site 1 and mScarlet at the C-terminus (Ppk1::pHluorinEC::mScarletC-term) in control neurons *(w1118)* and neurons lacking *ppk26 (11ppk26: ppk26/J111M1)*. The neurons are homozygous for ppk1:: pHluorinEC::mScarletC-term. Scale bar, 50 μm.
(PDF)

**S4 Fig. Characterization of a split-GFP-based approach to label membrane-expressed Ppk1.** Representative images of ddaC neurons in live 3rd instar larvae. (A) By itself, secGFP (1–10) is not fluorescent *(DcG-Ga/4 UAS-secGFP(1–10))* (top row). GFP(11) is not fluorescent in the absence of GFP(1–10) *(ppk1:: GFP(11x3)Ec::mCher,yG-1erm)(middle* row). The expression of GFP(1–10) in neurons does not result in fluorescence, consistent with the model that GFP(11), located on an extracellular loop of Ppk1, is not exposed to the cytoplasm where it might encounter GFP(1–10) *(ppk Ga/4 UAS-GFP(1–10)* in combination with *ppk1::GFP(11x3) Ec::mCher,yc-1erm;* the ddaC neurons also express CD4::tdTomato under the control of a *ppk* enhancer, *ppk-CD4::tdTomato)(bottom* row). (B) By itself, GFP(11) Ec:: CD4::tdTomato does not produce any GFP fluorescence (left). In the presence of secGFP(1–10), GFP(11)Ec::CD4:: tdTomato, fluorescent GFP is visible throughout the ddaC neuron (right). GFP(11) Ec:: CD4:: tdTomato is expressed in class IV da neurons under the control of a *ppk* enhancer, *ppk-GFP (11)Ec::CD4::tdTomato*. Black arrowheads point to the cell body. Scale bar, 50 μm.
(PDF)

**S5 Fig. Sholl analysis of dendritic arbors in dendrite growth mutants.** For each dendrite growth mutant, the data are plotted as mean± SEM; n represents number of neurons. *w1118* is the control genotype. Quantification, Sholl analysis (mean± SEM): Critical radius (μm), control = 127 ± 3, n = 12, and *Rp/22-RNAi* = 124 ± 6, n = 13, Student's unpaired t-test (p = 0.9467); control = 131 ± 4, n = 15, and *EcR-DN* = 138 ± 4, n = 15, Student's unpaired t-test (p = 0.2475); control = 146 ± 5, n = 15, and *Rac1 O/E* = 82 ± 4, n = 14, Student's unpaired t-test (p<0.0001); control = 132 ± 5, n = 12, and *P/3K O/E* = 130 ± 4, n = 12, Student's unpaired t-test (p = 0.7345). Maximum number of intersections, control = 38 ± 2, n = 12, and *Rp/22-RNAi* = 20 ± 1, n = 13, Student's unpaired t-test (p<0.0001); control = 41 ± 1, n = 15, and *EcR-DN* = 29 ± 1, n = 15, Student's unpaired t-test (p<0.0001); control = 42 ± 1, n = 15, and *Rac1 O/E* = 52 ± 2, n = 14, Student's unpaired t-test (p<0.0001); control = 47 ± 2, n = 12,

and *P/3K O/E* = 58 ± 2, n = 12, Student's unpaired t-test (p = 0.0002).
(PDF)

**S6 Fig. Effects of altering dendritic growth on GFP::TrpA1.** Representative images of ddaC neurons in live 3rd instar larvae. The dendritic membrane is marked by CD4::tdTomato. Dashed-outline boxes: Zoomed-in views of dendrite branches. Representative images of control larvae expressing endogenous TrpA1 tagged at the N-terminus with GFP (GFP::TrpA1) (top left). Representative image and quantification of EcR-DN-expressing neurons (top right). Quantification, GFP::TrpA1, dendrites: Student's unpaired t-test (p<0.0001); control (26 neurons, 11 larvae) v. EcR-DN (24 neurons, 10 larvae). Representative image and quantification of neurons over expressing Rac1 (bottom left). Quantification, GFP::TrpA1, dendrites: Student's unpaired t-test (p<0.0001); control (26 neurons, 11 larvae) v. Rac1 O/E (22 neurons, 11 larvae). The same control was used for both EcR-DN and Rac1 O/E (experiments were carried out in parallel). Control genotype: ppk-Gal4 ppk CD4::tdTomato; GFP::TrpA1. Experimental genotypes: ppk-Gal4 ppk-CD4::tdTomato/UAS-EcR-DN; GFP::TrpA1 and ppk-Gal4 ppk-CD4::tdTomato/UAS-Rac1; GFP::TrpA1. Scale bars, 50 μm and 10 μm (dashed outline boxes).
(PDF)

**S7 Fig. Ppk1 localizes to axons when dynein-mediated transport is disrupted.** Representative Images of ddaC neurons in live 3rd instar larvae. Dashed-outline boxes: Zoomed-in views of axons. (A) Representative images and quantification of Ppk1::sfGFP in control (21 neurons, 11 larvae) and *Dlic-RNAi*-expressing neurons (21 neurons, 11 larvae). Quantification, axons: Mann-Whitney test (p<0.0001). Scale bars, 50 μm and 5 μm (dashed-outline boxes). Representative images and quantification of Ppk1::sfGFP in control neurons and neurons over-expressing *dmn* (*O/E dmn*). Quantification, axons: Mann-Whitney test (p<0.0001); control (20 neurons, 12 larvae) v. *O/E dmn* (23 neurons, 14 larvae). Scale bars, 50 μm and 5 μm (dashed-outline boxes). (B) Representative images of Ppk1::sfGFP and GALNT2::TagRFP in the axons of control and *Dlic-RNAi*-expressing neurons. Zoomed in images of an axon of a *Dlic-RNAi*-expressing neuron shows that the Ppk1::sfGFP puncta (green arrowheads) do not colocalize with ectopic Golgi marked by GALNT2::TagRFP (magenta arrowheads). Scale bars, 10 μm and 5 μm (dashed-outline boxes). (C) Representative images of GALNT2::TagRFP and Ppk1::sfGFP in the axons of control neurons and neurons expressing a dominant-negative form of the Golgin Lava lamp (Lva) (Lva-DN); Lva-DN disrupts the interaction between Golgi and dynein but does not affect dynein activity. In neurons expressing *lva-DN*, GALNT2::TagRFP mislocalizes to axons (black arrowheads) and axonal Ppk1::sfGFP levels increase. Quantification, Ppk1::sfGFP, axons: Student's unpaired t-test (p<0.0001); control (17 neurons, 6 larvae) v. *lva-DN* (16 neurons, 7 larvae). Scale bar, 10 μm. Control genotypes: *w1118; ppk-Gal4*. Experimental genotypes: *w1118; ppk-Gal4 UAS-Dlic-RNAi UAS-Dicer* (A and B), *w1118; ppk-Gal4 UAS-dmn* (A), *w1118; ppk-Gal4 UAS-lva-DN* (C). *UAS-GALNT2::TagRFP* and *ppk1::sfGFP* included as indicated (A-C). In the graphs, each data point represents a neuron and data are plotted as mean ± SEM. ****p<0.0001. AU: arbitrary units.
(PDF)

**S8 Fig. Dynein implicated in transporting Ppk channels to dendrites.** Model illustrating the transport of Ppk channels from the Golgi to dendrites by dynein (left). Our data suggest that this post-Golgi transport involves Rab11, either directly or indirectly (e.g., Ppk channels may be transported to dendrites via Rab11(+) endosomes). When dynein activity is reduced after dendrites have initiated their growth (right), the supply of Ppk channels to dendrites decreases, which culminates in an increase of Ppk channels in the Golgi. Although the localization of Ppk channels to dendrites is reduced when dynein activity decreases, the density of Ppk channels

already in the dendritic membrane is not reduced. Created with Biorender.com.
(PDF)

**S1 Table. Primary data for FRAP experiment.**
(XLSX)

## Acknowledgments

We thank Drs. Yuh Nung Jan (University of California, San Francisco), Hsiu-Hsiang Lee (National Taiwan University College of Medicine), Bing Ye (University of Michigan), the Bloomington Drosophila Stock Center (NIH P40OD018537), and the Vienna Drosophila Resource Center for fly strains and antibodies. We thank Drs. Anjon Audhya and Jennifer Peotter (University of Wisconsin-Madison) for assistance with the Imaris software. We thank Wildonger lab members for their feedback and suggestions; in particular, we thank Jessica Liang for her contributions to developing the FRAP protocol, Dena Johnson-Schlitz for technical assistance and advice, and Dr. Harriet Saunders for helpful guidance. We thank the Biochemistry Department and Dr. Aaron Hoskins (University of Wisconsin-Madison) for generously supporting J.W.M..

## Author Contributions

**Conceptualization:** Josephine W. Mitchell, Jill Wildonger.

**Formal analysis:** Josephine W. Mitchell, Ipek Midillioglu, Ethan Schauer.

**Funding acquisition:** Chun Han, Jill Wildonger.

**Investigation:** Josephine W. Mitchell, Ipek Midillioglu, Ethan Schauer.

**Methodology:** Josephine W. Mitchell, Ipek Midillioglu, Ethan Schauer.

**Project administration:** Jill Wildonger.

**Resources:** Bei Wang, Chun Han.

**Supervision:** Jill Wildonger.

**Validation:** Josephine W. Mitchell, Ipek Midillioglu, Ethan Schauer.

**Visualization:** Josephine W. Mitchell, Ipek Midillioglu, Ethan Schauer, Jill Wildonger.

**Writing – original draft:** Josephine W. Mitchell, Jill Wildonger.

**Writing – review & editing:** Josephine W. Mitchell, Ipek Midillioglu, Ethan Schauer, Bei Wang, Chun Han, Jill Wildonger.

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
