## [Decision Letter · Decision Letter 0]

22 Sep 2023

Dear Dr Wildonger,

Thank you very much for submitting your Research Article entitled 'Coordination of Pickpocket ion channel delivery and dendrite growth in Drosophila sensory neurons' to PLOS Genetics.

The manuscript was fully evaluated at the editorial level and by independent peer reviewers. The reviewers appreciated the attention to an important topic but identified some concerns that we ask you address in a revised manuscript.

We therefore ask you to modify the manuscript according to the review recommendations. Your revisions should address the specific points made by each reviewer.

Yours sincerely,

Gaiti Hasan

Academic Editor

PLOS Genetics

Gregory P. Copenhaver

Editor-in-Chief

PLOS Genetics

Reviewer's Responses to Questions

**Comments to the Authors:**

Reviewer #1: The authors have partially addressed my previous concerns. I still wonder whether the coordinated distribution of Ppk1 with dendritic arborization is a functional scaling or not, but I also understand the technical difficulties to answer my previous comments. Thus, I have no further comments to the submitted manuscript.

Reviewer #2: The authors have generated tools to study the mechanisms regulating the trafficking of Ppk1 into dendrites of DDa neurons by generating different tagged versions of the protein. They have tested these tools rigorously to demonstrate that they can be used reliably to monitor localization and trafficking of Ppk1. Using these reagents the authors show that

Ppk1 channel are present from the very early stages of dendtrite formation. The levels of these channels (normalized to membrane area) is relatively constant and does not change in response to altered dendrite morphology; these channels do not diffuse rapidly in the membrane. Trafficking to the dendrites is via golgi and golgi outposts and is dependent on Rab11. Loss of Rab11 leads to decrease in channel levels at the membrane. Ppk1 is required for membrane localization of Ppk26DEG: loss of Ppk1 enhances the arborization phenotype of Ppk26DEG. This is an improved version of the manuscript. The concerns raised earlier have been addressed. The authors have also developed and used an improved version of the split GFP system in their study.

(i) One major concern here is the opposing results seen upon perturbing dynein function. The authors show that disrupting dynein activity does not have an immediate effect on Ppk levels in the membrane. But, over time (in a 3rd instar larva) there is a decrease in channel levels measured using Ppk1-sfGFP levels.

However, they see the opposite with the split GFP system ie. there is an accumulation of Ppk1 in the membrane. Interestingly, immunostaining using non-detergent conditions shows no change. The authors suggest that the increase in intensity observed with the split GFP system compared with immunostaining is likely due to increased sensitivity. However, they do not explain why the result is opposite to what is seen with Ppk1-sfGFP. In the discussion they state that the increase in intensity seen with the split-GFP system is likely due to poor turnover of the Ppk1 protein and they use the slow diffusion rates seen in FRAP to support this conclusion. However, this argument is rather weak. Not sure if FRAP can be used to comment on protein stability at the membrane.

Minor concern

(ii) Lines 372-373: ‘Our results also suggest that while overall levels of Ppk1 decrease when 373 dynein function is perturbed over time (e.g., Ppk1::GFP; Fig 6A), Ppk1 that does make it into 374 dendrites and the dendritic membrane is not diminished by reduced dynein activity.’ Not sure what is meant here.

Reviewer #3: Authors have significantly improved the manuscript.

The study attempts to understand the trafficking of the sensory neuron ion channel in one class of Drosophila sensory neurons. The strength is the manuscript is the nice split-GFP system. The weakness is the lack of any real mechanistic insight. However, the study is well worth publishing, is well done and significantly improved since the earlier submission. Several things of note: very few systematic studies that address trafficking of an endogenous channel in vivo which this study does, the assessment of the role of ppk26 (old problem how do multiple proteins of the same complex help in targeting), phenotypic roles in development of the dendritic arbor.

I have a few minor suggestions which I hope will improve the manuscript.

It would be good if authors mention & discuss the possibility that "mis-localization of Golgi alone can affect axonal density and distribution of Ppk1" that their data suggest and they themselves say in the reviewer response document. I also think it would be useful to add the Sholl analysis in the supplement given this analysis method is widely used. Finally, a diagram of how authors envisage this trafficking works would not be amiss.

**Have all data underlying the figures and results presented in the manuscript been provided?**

Reviewer #1: Yes

Reviewer #2: Yes

Reviewer #3: **No: **I think excel sheets of some of the data should ideally be provided. I was not able to find it.

PLOS authors have the option to publish the peer review history of their article (what does this mean?). If published, this will include your full peer review and any attached files.

Reviewer #1: No

Reviewer #2: No

Reviewer #3: No

---

## [Decision Letter · Decision Letter 1]

23 Oct 2023

Dear Dr Wildonger,

We are pleased to inform you that your manuscript entitled "Coordination of Pickpocket ion channel delivery and dendrite growth in Drosophila sensory neurons" has been editorially accepted for publication in PLOS Genetics. Congratulations!

Yours sincerely,

Gaiti Hasan

Academic Editor

PLOS Genetics

Gregory P. Copenhaver

Editor-in-Chief

PLOS Genetics

Comments from the reviewers (if applicable):

Reviewer's Responses to Questions

**Comments to the Authors:**

Reviewer #2: The authors have answered all my queries.

**Have all data underlying the figures and results presented in the manuscript been provided?**

Reviewer #2: None

PLOS authors have the option to publish the peer review history of their article (what does this mean?). If published, this will include your full peer review and any attached files.

Reviewer #2: No

**Data Deposition**

http://datadryad.org/submit?journalID=pgenetics&manu=PGENETICS-D-23-00866R1

**Press Queries**

---

## [Editor Report · Acceptance letter]

4 Nov 2023

PGENETICS-D-23-00866R1 

Coordination of Pickpocket ion channel delivery and dendrite growth in Drosophila sensory neurons 

Dear Dr Wildonger, 

We are pleased to inform you that your manuscript entitled "Coordination of Pickpocket ion channel delivery and dendrite growth in Drosophila sensory neurons" has been formally accepted for publication in PLOS Genetics! Your manuscript is now with our production department and you will be notified of the publication date in due course.

With kind regards,

Lilla Horvath

PLOS Genetics

On behalf of:
